# Adaptively Coordinating with Novel Partners via Learned Latent Strategies

**Benjamin Li**[1]    **Shuyang Shi**[1]    **Lucia Romero**[2]    **Huao Li**[2]    **Yaqi Xie**[1]    **Woojun Kim**[1]
**Stefanos Nikolaidis**[3]    **Michael Lewis**[2]    **Katia Sycara**[1]    **Simon Stepputtis**[4]

[1]Carnegie Mellon University    [2]University of Pittsburgh
[3]University of Southern California    [4]Virginia Tech
{benjil, shuyangs, yaqix, woojunk, sycara}@andrew.cmu.edu
{luciaromero, hul52, cmlewis}@pitt.edu
nikolaid@usc.edu, stepputtis@vt.edu

## Abstract

Adaptation is the cornerstone of effective collaboration among heterogeneous team members. In human-agent teams, artificial agents need to adapt to their human partners in real time, as individuals often have unique preferences and policies that may change dynamically throughout interactions. This becomes particularly challenging in tasks with time pressure and complex strategic spaces, where identifying partner behaviors and selecting suitable responses is difficult. In this work, we introduce a strategy-conditioned cooperator framework that learns to represent, categorize, and adapt to a broad range of potential partner strategies in real-time. Our approach encodes strategies with a variational autoencoder to learn a latent strategy space from agent trajectory data, identifies distinct strategy types through clustering, and trains a cooperator agent conditioned on these clusters by generating partners of each strategy type. For online adaptation to novel partners, we leverage a fixed-share regret minimization algorithm that dynamically infers and adjusts the partner's strategy estimation during interaction. We evaluate our method in a modified version of the Overcooked domain, a complex collaborative cooking environment that requires effective coordination among two players with a diverse potential strategy space. Through these experiments and an online user study, we demonstrate that our proposed agent achieves state of the art performance compared to existing baselines when paired with novel human, and agent teammates.

## 1   Introduction

As an increasing number of AI agents and robots enter our daily lives, it is becoming increasingly important to explore methods of effective collaboration between agent and human. When an agent attempts to collaborate with an unknown partner (whether human or artificial), the challenge can typically be divided into two steps: (1) accurately predicting the partner's behavior, and (2) choosing actions that advance the team toward its common goal. Our approach focuses on understanding and responding to behavioral patterns that distinguish different types of collaborators. Equipping an agent with the ability to identify partner type can enable it to exploit knowledge about that type when choosing its own actions, enabling adaptation for coherent and fluid collaboration. This type of adaptation is essential, as misaligned actions between teammates inevitably decrease overall performance. The problem of adapting to a previously unknown human is an instance of the more general problem of ad hoc teamwork, cooperating with any previously unknown teammate [38]. The difficulties of the problem arise if the agent begins without prior knowledge of its teammate's behavioral patterns, yet must select its own actions to match the teammate's intended strategy

39th Conference on Neural Information Processing Systems (NeurIPS 2025).

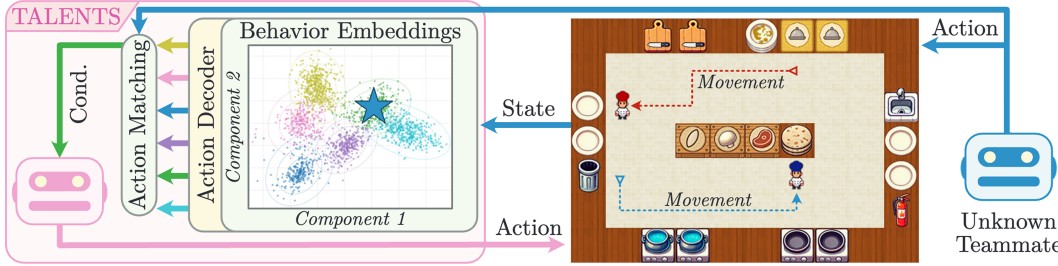

Figure 1: Overview of TALENTS: Provided an observation of a teammate, the VAE's latent strategy clusters are queried to generate action predictions. At subsequent timesteps, the teammate's actual actions are compared to these predictions, and the belief over the teammate's latent strategy is progressively updated.

for completing the collective goal. This presents significant challenges given the multiplicity of possible strategies for any collaborative task. The difficulty intensifies in human-agent partnerships, where humans typically employ diverse, non-stationary, and often suboptimal approaches that can shift rapidly during interaction. Consequently, effective collaborative agents must be capable of interpreting their partners' intentions and behaviors in real-time while demonstrating rapid adaptive responses.

In coordinated team settings, roles and specializations can be learned by agents in a centralized manner [23; 43]. However, human-agent teams are often *decentralized*, in that roles and strategies are often implicit and need to be inferred [24]. To account for the diverse set of teammates an agent may encounter at test-time, an effective cooperator agent in this setting must be able to successfully learn and play the best response to each potential teammate's strategy. A popular class of methods that seeks to address this is population-based training, in which a population of agent policies is generated, from which agents are drawn to serve as teammates to train a cooperator [18; 41]. At their core, these methods seek to construct agent populations with sufficient behavioral diversity to span the distribution of strategies exhibited by human players, often aided by a secondary objective during population training that enforces diversity in the set of training partners [4; 25; 34; 39; 51]. To further increase the strategic diversity of training partners, these agent populations can be used to train a generative model, which enables sampling from a continuous latent strategy distribution to simulate a much wider range of novel teammates [22]. Complementary to population-based approaches, agent modeling methods focus on online adaptation by inferring teammate behaviors and conditioning agent actions on these inferred strategies [29; 46; 50; 52]. However, these approaches often rely on a limited or hand-defined collection of strategy types to operate over, or lack the diverse synthetic training data provided by population-based methods. We argue that unifying these two paradigms can enable more effective team collaboration: leveraging diverse population data to learn a latent strategy space from which unique partners can be generated during training, while explicitly reasoning over teammate types at test-time to enable rapid online adaptation and best-response action selection.

In this work, we introduce **T**eam **A**daptation via **L**at**E**nt **N**o-regre**T** **S**trategies (**TALENTS**) (see Fig. 1), a novel method for zero-shot coordination that models partner behaviors in a semantic strategy space. At the core of our **TALENTS** agent is a latent behavior embedding created by a variational autoencoder. This autoencoder is trained from offline trajectory data in the respective environment, collected from a population of baseline agents. With this dataset, we train a variational auto-encoder (VAE) with the goal of predicting each agent's next action and utilize its latent space as behavior representation. Next, we determine an appropriate number and location of behavior clusters utilizing K-Means with silhouette analysis [32]. At test time, the type of a new partner, with respect to the behavior clusters, is determined by assessing each latent cluster mean's decoded prediction accuracy with respect to the partner's actual action (observed at the next timestep). Our agent's belief of its partner is updated according to this per-step loss, and determines the most likely partner type which is then used as a conditioning term for its own action.

We evaluate our agent in a modified Overcooked environment that extends prior work [2; 10; 45] with additional temporal pressure through order timers and bonus rewards for fast deliveries. This setting requires the coordinated use of three cooking stations to prepare two distinct recipes, significantly

increasing the demands for effective teamwork and strategic coordination. We conduct experiments in both agent-agent and human-agent zero-shot coordination settings. In agent-agent evaluations, our method demonstrates overall superior performance compared to existing baselines. Furthermore, we deploy these same agents in human studies, demonstrating their ability to collaborate effectively with diverse unseen human partners. In summary, the work introduces the following contributions:

- We introduce a novel method for learning strategy-specific best responses for human-agent collaboration using a generative model that characterizes agents based on latent strategies.

- Our method learns a strategy-conditioned cooperator that assesses teammate type and adapts its behavior through an online regret-minimization framework.

- In extensive teaming experiments, including a human-agent subject study, we show that our agent achieves state-of-the-art performance in a challenging Overcooked environment, adapting to new partners in a similar strategy space in a zero-shot manner.

## 2    Related Work

**Zero-Shot Coordination**    Collaborating with partners that were unseen by agents during training is the central goal of Zero-Shot Coordination (ZSC) [14; 38]. Self-play (SP) is a common approach to address this problem, in which agents establish a joint strategy by pairing with a copy of itself during training [33].These approaches have been shown to surpass human performance in zero-sum games in which agents can exploit sub-optimalities of opponents at test time [1; 37], but are often unable to achieve comparable performance in common-payoff cooperative settings since these agents tend to develop rigid behavior patterns [2]. Population-based training (PBT) methods combat this rigid convention formation by optimizing over diverse partner sets [18; 41]. Fictitious Co-Play (FCP) [39] leverages intermediate checkpoints of partner policies to simulate teammates of different skill levels, while other methods increase the strategy coverage of the population by means of diversity or entropy related objectives [25; 30; 51], cross-play performance [4; 34], or reward shaping [40; 44; 48]. E3T [47] applies the entropy objectives of PBT methods to the more efficient SP training paradigm, defining the trainer partner as a mixture between the ego policy (maximizing coordination) and a random policy (maximizing entropy). GAMMA [22] further enhances the breadth of PBT strategy coverage by learning a generative model over the population of policies, encoding trajectories into latent representations. They are then able to perform targeted sampling over these latents to generate novel agents spanning the strategy space of the original population. Our proposed TALENTS method applies the same approach of GAMMA by training a generative model from PBT agent trajectory rollouts, but differs in that we then cluster the learned latent space and perform controlled sampling over the clusters. We treat these clusters as a population of strategy types, training a cooperator agent conditioned on these clusters to explicitly learn role-compatible behaviors.

**Strategy Inference**    Inferring the partner type allows agents to form beliefs about other agents in the environment and optimize its own behavior in response to those beliefs, whether in adversarial or cooperative environments. A common strategy is to learn latent embeddings of partner type, role, or skill from a dataset of trajectories [13; 28; 36; 46; 42]. Other methods directly estimate the opponent's policy updates [6], perform Bayesian inference of partner type [3; 7; 9; 15; 52], or learn a partner model through interaction during training [16; 26; 47]. In MeLIBA [52], a Bayesian filtering objective is used to model confidence in the partner's type as the trajectory rolls out, allowing a best response agent to respond optimally given its knowledge about its opponent. Grover et al. [11] proposes a triplet loss term to encourage distinguishability between different types of opponents, and Papoudakis and Albrecht [29] expands on this by using reinforcement learning to train an agent conditioned on this latent embedding. Zhao et al. [50] use apprenticeship learning to model the different types of observed strategies, then online mixture of experts to select compatible policies over the course of an episode. Li et al. [19] trains a library of teammate types, each with a different role that, when faced with an unknown human, use cross-entropy to choose strategies that are highly compatible with the human's inferred current strategy. A separate but adjacent line of work learns Theory of Mind models to predict partner preferences, intentions, or beliefs [20; 21; 31; 49]. Our work proposes to use the same latent space that generated training partners to perform strategy inference. At train-time, the cooperator agent is conditioned on the latent cluster its teammate is generated from in order to learn strategy-specific best responses. At test-time, we utilize a tracking-regret minimized approach

to enable *intra*-episodic adaptation, improving our agent's ability when paired with teammates that may change their strategy several times over the course of a single episode.

# 3 Preliminaries

In this section, we establish the mathematical framework for our multi-agent reinforcement learning setup conditioned on latent strategies.

## 3.1 Two-Player Markov Decision Process

A Markov Decision Process (MDP) is defined by a tuple $\mathcal{M} = (\mathcal{S}, \mathcal{A}_1, \mathcal{A}_2, P, R, \gamma, H)$. While we present the two-agent formulation for clarity, this framework naturally extends to an arbitrary number of agents. In this formulation, $(\mathcal{S})$ represents a set of states, and $(\mathcal{A}_i)$ denotes the set of actions available to agent (i), where $(i \in 1, 2)$. The transition probability function $P : \mathcal{S} \times \mathcal{A}_1 \times \mathcal{A}_2 \times \mathcal{S} \to [0, 1]$ defines the probability $P(s'|s, a)$ of transitioning to state $s'$ where a joint action $a$ is taken in state $s$. The team reward function $R : \mathcal{S} \times \mathcal{A}_1 \times \mathcal{A}_2 \times \mathcal{S} \to \mathbb{R})$ specifies the reward $R(s, a_1, a_2, s')$ received by both agents when transitioning from state $s$ to state $s'$ after executing the joint actions $a_1$ and $a_2$. The discount factor $\gamma \in [0, 1)$ determines the significance of future rewards, influencing how current decisions are evaluated relative to potential long-term outcomes. Lastly, $H$ represents the horizon of the problem, which is typically omitted in infinite horizon problems.

The goal in a multi-agent MDP is to find a policy $\pi : \mathcal{S} \to \mathcal{A}_1, \mathcal{A}_2$ that maximizes the expected discounted sum of rewards, defined as $\mathbb{E}_\pi[\sum_{t=0}^{\infty} \gamma^t R(s_t, a_{1,t}, a_{2,t}, s_{t+1})]$.

## 3.2 Hidden Parameter Markov Decision Process

A Hidden Parameter Markov Decision Process (HiP-MDP) extends the standard MDP framework to incorporate unobserved parameters that influence the transition and reward dynamics. Formally, a HiP-MDP is defined as a tuple $\mathcal{M}(\mathcal{Z}) = (\mathcal{S}, \mathcal{A}_1, \mathcal{A}_2, \mathcal{Z}, P, P^z, R_1, R_2, \gamma, H)$ where $\mathcal{S}, \mathcal{A}, \gamma$, and $H$ are defined as in the standard MDP. $\mathcal{Z}$ is a set of hidden parameters and $P : \mathcal{S} \times \mathcal{A}_1 \times \mathcal{A}_2 \times \mathcal{S} \times \mathcal{Z} \to [0, 1]$ is a transition probability function that depends on the hidden parameter $z \in \mathcal{Z}$. $P^z : \mathcal{S} \times \mathcal{A}_1 \times \mathcal{A}_2 \times \mathcal{S} \times \mathcal{Z}$ is a transition probability function for the hidden parameter $z \in \mathcal{Z}$, representing how $z$ may evolve over the course of an episode. $R$ is defined as $\mathcal{S} \times \mathcal{A}_1 \times \mathcal{A}_2 \times \mathcal{S} \times \mathcal{Z} \to \mathbb{R}$, representing the reward function that depends on the hidden parameter $z \in \mathcal{Z}$.

For a specific, constant value of the hidden parameter $z \in \mathcal{Z}$, the HiP-MDP reduces to a standard MDP $\mathcal{M}_z = (\mathcal{S}, \mathcal{A}_1, \mathcal{A}_2, P(\cdot|z), R(\cdot|z), \gamma, H)$. Given the two-player HiP-MDP framework, the learning objective for each agent $i$ can be formulated as:

$$\max_{\pi_i} \mathbb{E}_{\pi_1, \pi_2, z}[\sum_{t=0}^{\infty} \gamma^t R_i(s_t, a_{1,t}, a_{2,t}, s_{t+1}, z)] \tag{1}$$

## 3.3 Tracking Regret Minimization

To address the strategic adaptations and non-stationarity inherent in our HiP-MDP setting, where teammates may shift between different strategies during an episode, we incorporate tracking regret minimization. This permits a limited number of optimal response policy changes over the course of an episode while maintaining performance guarantees. In online learning, at each time step $t$, an agent selects a decision $x_t$ from a convex set $\mathcal{X}$ and subsequently observes a loss function $f_t : \mathcal{X} \to \mathbb{R}$. Unlike static regret, which compares against a single fixed decision, tracking regret compares the agent's performance against a sequence of decisions that can change a limited number of times.

Formally, for a partition $\mathcal{I} = \{I_1, I_2, \ldots, I_m\}$ of the time horizon $[T]$ into $m$ intervals, and a sequence of comparators $\mathbf{u} = (u_1, u_2, \ldots, u_m)$ where each $u_j \in \mathcal{X}$, the tracking regret is defined as:

$$Reg_T^s(\mathcal{I}, \mathbf{u}) = \sum_{t=1}^{T} f_t(x_t) - \sum_{j=1}^{m} \sum_{t \in I_j} f_t(u_j). \tag{2}$$

The worst-case tracking regret against any sequence with at most $m$ expert policies (i.e. $m-1$ expert changes) is

$$Reg_T^s(m) = \max_{\mathcal{I}, \mathbf{u}} Reg_T^s(\mathcal{I}, \mathbf{u}).$$ (3)

Fixed Share [12] with optimally-selected switching rate $\alpha$ achieves the following bound for any comparator sequence of $m$ segments:

$$Reg_T^s(m) = O\left(\sqrt{T\left(m \ln N + m \ln \frac{T}{m}\right)}\right).$$ (4)

# 4 Team Adaptation via No-Regret Strategies

We introduce TALENTS, a method of training a general cooperator agent that is conditioned on a wide range of strategies and behaviors. In the online setting, the cooperator agent can assess the strategy of its partner and adapt to it by playing the corresponding best response policy. Given a dataset of offline trajectories exhibiting a wide range of strategies, we first use a variational autoencoder (VAE) [17] to learn a latent space of strategy vectors. We then partition the latent space into discrete clusters, with each cluster representing a unique high-level strategy. Next, we train a best response agent over the different strategy clusters, generating a strategic partner agent for every episode by selecting a strategy cluster and sampling from its latent mean. We condition the cooperator agent on its training partner's mean in order to learn a strategy-dependent best response. At test time, we use the fixed share algorithm to select a latent mean to condition our cooperator agent on, allowing intra-episodic adaptation to the new partner.

## 4.1 Strategy Learning

To form the latent strategy space, we utilize an offline dataset of trajectories generated by agent-agent rollouts. Note that this data can alternatively come from human gameplay [13; 50], or a mix of human and agent data [22]. However, this work focuses on exploring ad hoc collaboration with humans without seeing human data at train time. The dataset, $\mathcal{D}_{traj} = \{\tau_i\}_{i=1}^N$, is composed of observation-action pairs for each agent collected from a pre-trained set of population agents, $\{o_t^{(i)}, a_t^{(i)}\}_{t=0}^T, i \in \{1, 2\}$. This dataset contains a diverse set of agent strategies and behaviors increasing the amount of covered latent space over the set of possible agent types. With this dataset, we subsequently train a VAE to approximate the posterior distribution of the trajectories. The VAE architecture consists of an encoder network $q_\phi(z|\tau)$ that maps trajectories $\tau$ to a distribution over latent strategy variables $z$, and a sequential decoder network $p_\theta(a_{t:t+H}|z, o_t)$ that predicts the next $H$ actions given the latent variable and current observation, insight that follows from Zintgraf et al. [52]. Specifically, the encoder parameterized by $\phi$ compresses the trajectory information into a low-dimensional latent space, producing parameters of a multivariate Gaussian distribution $\mathcal{N}(\mu_\phi(\tau), \Sigma_\phi(\tau))$. By predicting the $H$ future actions of the agent, we learn a representation of the agent's long-term intent. Importantly, we will later be able to leverage the decoder as a method for generating agents of specific behavior types.

The objective of the sequential VAE is to maximize the log-likelihood of the future actions given past observations and actions: $\log p_\theta(a_{t:t+H}|o_t, \tau_{t-h:t})$. However, directly computing this is intractable as it requires marginalizing over all possible latent variables: $\log p_\theta(a_{t:t+H}|o_t, \tau_{t-h:t}) = \log \int p_\theta(a_{t:t+H}|z, o_t)p(z|\tau_{t-h:t})dz$. Instead, we optimize the Evidence Lower Bound (ELBO):

$$\mathcal{L}(\theta, \phi; \tau) = \mathbb{E}_{z \sim q_\phi(z|\tau_{t-h:t})}[\log p_\theta(a_{t:t+H}|z, o_t)] - \beta D_{KL}(q_\phi(z|\tau_{t-h:t})||p(z))$$ (5)

Once we have learned our VAE, we cluster the latent space. To this end, we perform K-means clustering with silhouette analysis [32] to determine the optimal clusters and the number of clusters in an unsupervised fashion. Each cluster represents a unique agent type, with which we utilize to train our best response cooperator agent.

## 4.2 Learning a Strategy-Conditioned Cooperator Agent

To train our agent, we utilize the strategy clusters and leverage the generative capability of our trained VAE. We retrieve actions for each agent type by sampling from the latent mean of each strategy cluster

and decoding them, conditioned on the observed environment state. For each episode in the training process, we randomly select one such cluster to sample from, using the priority-based sampling technique proposed in Zhao et al. [51]. Using a categorical variable corresponding to the partner cluster, we learn a bias vector corresponding to the cooperator's action space. The actor network's output logits are then biased by this vector, explicitly encouraging or dissuading the cooperator from taking certain actions depending on the partner type (see algorithm 1). This motivates the cooperator to learn the unique conventions needed to best respond to its partner (Note that we utilize a high-level action space in our experiments). We train our cooperator agent using independent PPO [5; 35].

---

**Algorithm 1** Strategy-Conditioned Cooperator Training

1: **Input:** Trained VAE encoder $q_\phi(z|\tau)$, decoder $p_\theta(a_{t:t+H}|z,o_t)$, strategy clusters $\{\mathcal{N}(\mu_1, \sigma_1^2), \mathcal{N}(\mu_2, \sigma_2^2), \dots, \mathcal{N}(\mu_K, \sigma_K^2)\}$, priority distribution $p(c)$
2: Initialize cooperator policy $\pi_\theta(\cdot|o, c)$
3: **for** each training episode **do**
4:      Sample a strategy cluster $c \sim p(c)$
5:      Sample latent strategy $z \sim \mathcal{N}(\mu_c, \sigma_c^2 I)$
6:      **for** each timestep $t$ in episode **do**
7:          Observe state $o_t$ for cooperator agent
8:          Compute action bias vector $b_c$ from cluster embedding matrix $E$, where $b_c = E[c]$
9:          Compute action logits from actor network $l_t = f_\theta(o_t)$
10:         Apply bias to logits: $\tilde{l}_t = l_t + b_c$
11:         Sample action from biased distribution $a_t \sim \text{softmax}(\tilde{l}_t)$
12:         Sample generated action for partner $a_{c,t} \sim p_\theta(\cdot \mid z, o_t)$
13:         Execute actions and collect rewards
14:     **end for**
15:     Update $\pi_\theta$ using PPO with collected trajectory
16:     Update priority-based sampling weights using total episodic return
17: **end for**
18: **Return:** Strategy-conditioned cooperator policy $\pi_\theta$

---

### 4.3   Online Adaptation to Novel Partners

The trained cooperator agent will be able to best respond to any partner strategy in the VAE distribution, provided that it has knowledge on which latent strategy the partner is playing. This adds the additional challenge of needing to infer the partner's type in the online setting. An approach that has been explored in previous research is to encode the novel partner's trajectory segment into the latent space and conditioning the cooperator on that [13; 46], but in practice distribution shift between train-time and test-time trajectories can lead to brittleness or suboptimal encoded representations, particularly when playing with humans. Instead, we utilize regret minimization and a no-regret algorithm to represent the partner's behavior.

We implement a variant of the fixed-share algorithm to infer the partner type, setting each strategy cluster as an "expert". At test time, we sample a latent representation from each cluster and decode into the predicted action, conditioned on the current observation. At the next timestep, we calculate the incurred loss by comparing the previous timestep's prediction with the actual action of the partner, then update the weights of each expert accordingly. Importantly, the fixed-share algorithm differs from standard static no-regret methods (such as Hedge [8], FTRL [27]), in that it minimizes the regret given that the expert may switch policies $m - 1$ times during an episode (algorithm 2). This is to account for the fact that policies at test-time may be non-stationary and might change latent strategy $z$ over the course of the episode, due to influence from its partner or continuous adaptation and learning from the task. We examine the effect of fixed-share versus static-regret minimization in Figure 4.

Fixed-share allows the agent to model these strategy changes and adapt its behavior during the episode. It also provides the regret bound that if the agent encounters a truly novel strategy at test time, the agent will only perform as poorly as the best-response to the closest observed strategy during training.

**Algorithm 2** Online Adaptation to Novel Partners with Fixed-Share

---

1: **Input:** Latent clusters $\{\mathcal{N}(\mu_1, \sigma_1^2), \mathcal{N}(\mu_2, \sigma_2^2), \ldots, \mathcal{N}(\mu_K, \sigma_K^2)\}$, trained cooperator policy $\pi_\theta(\cdot|o, c)$, VAE decoder $p_\theta$, switching parameter $\alpha \in (0, 1)$, learning rate $\eta > 0$
2: Initialize weight vector $w^1 = (1/K, \ldots, 1/K)$ uniformly over $K$ experts
3: **for** $t = 1, 2, \ldots$ **do**
4:     Observe current state $o_t$
5:     **for** each cluster $c \in \{1, 2, \ldots, K\}$ **do**
6:         Sample latent strategy $z_c \sim \mathcal{N}(\mu_c, \sigma_c^2 I)$
7:         Predict partner action $\hat{a}_t^c = \arg\max_a p_\theta(a|z_c, o_t)$
8:     **end for**
9:     Compute leading expert $c^* = \arg\max_c w_c^t$
10:    Execute cooperator action $a_t \sim \pi_\theta(\cdot|o_t, c^*)$
11:    Observe partner's actual action $a_t^p$
12:    Compute loss for each expert: $\ell_c^t = -\log p_\theta(a_t^p|z_c, o_t)$
13:    Update pre-sharing weights: $\tilde{w}_c^{t+1} = w_c^t \exp(-\eta \ell_c^t)$
14:    Normalize: $\tilde{w}^{t+1} = \tilde{w}^{t+1} / \sum_{c=1}^K \tilde{w}_c^{t+1}$
15:    Apply fixed-share update: $w_c^{t+1} = (1-\alpha)\tilde{w}_c^{t+1} + \alpha \sum_{j=1}^K \tilde{w}_j^{t+1}/K$
16: **end for**

---

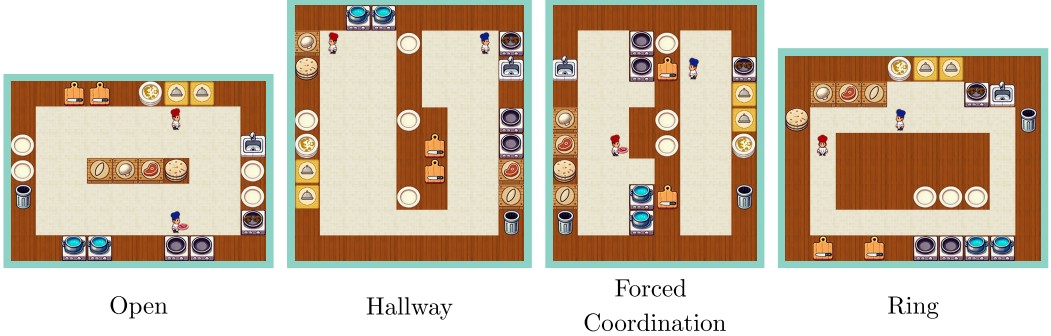

| Open | Hallway | Forced Coordination | Ring |

Figure 2: The four Overcooked layouts used in experiments.

## 5 Experiments

We evaluate our algorithm in a modified version of the Overcooked-ai environment [2] across agent-agent and human-agent gameplay. In this domain, we evaluate across four layouts (see Fig. 2) for agent-agent teams, and three for human-agent teams. These layouts are conceptually similar to those proposed in [2], but feature additional task complexity and timing constraints to further highlight the need for effective teamwork.

### 5.1 Agent-Agent Zero-Shot Coordination

We evaluate TALENTS in agent-agent settings using three different agent populations: Fictitious Co-Play (FCP) [39], Maximum Entropy Population (MEP) [51], and Behavior Preference (BP) Agents [44]. For the FCP and MEP populations, eight self-play agents are initialized with different seeds and trained. The MEP population is trained with an additional population entropy loss [51]. The final FCP and MEP populations consist of the final policies as well as two intermediary policy checkpoints per policy. As stated by Wang et al. [44], a well-designed population of behavior preference presents behavior diversity that is more similar to that of human teams, compared to other population generation methods. Therefore, we select it as another training population and train 24 behavior preference (BP) agents to cover a wide spectrum of partner strategies. The BP is represented by a linear combination of event-based shaped reward terms that encourage the agent to learn various behaviors. Using these populations, we compare our method to two baselines: the population-trained best response cooperator (**BR**), and a **GAMMA** cooperator. Notably, we select **GAMMA** as a

Table 1: Resulting scores of games played with a held-out set of 12 behavior-preferenced SP agents (mean ± standard error).

| Pop | Agent | Open | Hallway | Forced-Coord | Ring |
|---|---|---|---|---|---|
| **FCP** | TALENTS | **710.36 ± 88.75** | **635.59 ± 107.54** | 34.38 ± 6.59 | **596.19 ± 33.34** |
| | GAMMA | 616.67 ± 14.99 | 537.60 ± 26.75 | 38.36 ± 6.65 | 395.03 ± 10.09 |
| | BR | 427.07 ± 14.17 | 366.14 ± 70.50 | **56.09 ± 12.33** | 288.61 ± 31.85 |
| **MEP** | TALENTS | **720.84 ± 72.63** | **640.51 ± 27.36** | 36.53 ± 2.58 | **568.11 ± 8.99** |
| | GAMMA | 682.38 ± 14.99 | 575.23 ± 21.17 | 31.89 ± 4.33 | 369.15 ± 12.47 |
| | BR | 437.00 ± 27.76 | 440.48 ± 113.03 | **50.40 ± 14.29** | 335.61 ± 28.48 |
| **BP** | TALENTS | **842.39 ± 36.47** | **642.94 ± 81.00** | 56.55 ± 9.09 | **647.62 ± 16.96** |
| | GAMMA | 573.93 ± 52.69 | 513.47 ± 34.06 | 47.52 ± 2.61 | 387.58 ± 17.41 |
| | BR | 469.88 ± 52.01 | 454.26 ± 86.5 | **78.53 ± 8.55** | 640.43 ± 28.06 |

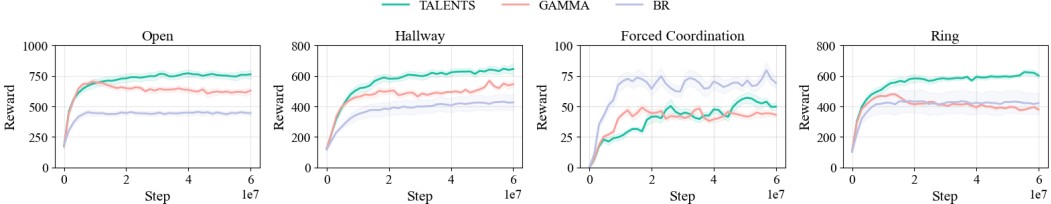

Figure 3: Evaluation performance during training for each layout, averaged across agents trained with FCP, MEP, and BP populations.

baseline due to the method also leveraging a generative model in its cooperator training process. Through our experiments, we wish to evaluate if our strategy-specific agent generation and cooperator conditioning will result in a more robust adaptive cooperator agent.

To generate the trajectories for VAE training, we select agents from the population to form a team and perform several joint rollouts, repeating until joint trajectories between all the agents in the population is achieved. This set of trajectories is used to train VAEs for both TALENTS and **GAMMA**.

To compare the performance of the methods with unseen agent partners, we evaluate using a held-out set of behavior preference agents representing a diverse span of strategies and competencies (see Tab. 1). We find that with the exception of the *Forced Coordination* layout, TALENTS is able to achieve the highest reward out of the evaluated methods, regardless of the population it was trained on. We hypothesize that both TALENTS and **GAMMA** are less effective than the population **BR** agent in *Forced Coordination* due to there being a very clear partition of duties between the teammates. **GAMMA** and TALENTS, which focus on exploring the diverse space of possible partners (many of which exhibit ineffective or uncooperative behavior), receive less reward signal when paired with agents of lower skill and are, as a result of the sparser training, less proficient in the overall task than the **BR** agent.

To evaluate the efficacy of the tracking-regret minimization framework, we ablate TALENTS by replacing the fixed-share algorithm with a static-regret minimizer. We use exponential weights to keep the weight update structure constant, only ablating the weight-sharing component. We then evaluate our cooperator with a partner policy randomly selected from a held-out behavior preference set, but replace the partner agent with a policy with a different behavior preference halfway through the episode. We see in Fig. 4 that the static-regret ablation is unable to update its belief to the new partner, and suffers lowered reward in the latter half of the episode as a result.

All models were trained on a server with 2× AMD EPYC 7713 64-core processors, 1.08 TB of system memory, and 5× Nvidia RTX 6000 Ada GPUs. In our experiments, VAE train time is typically 4-6 hours, while a cooperator agent can be trained in approximately 24 hours.

## 5.2 Human-Agent Zero-Shot Coordination

In this section, we evaluate our proposed method against state-of-the-art multi-agent coordination baselines with real human participants. Data collection is done on online crowd-sourcing platforms

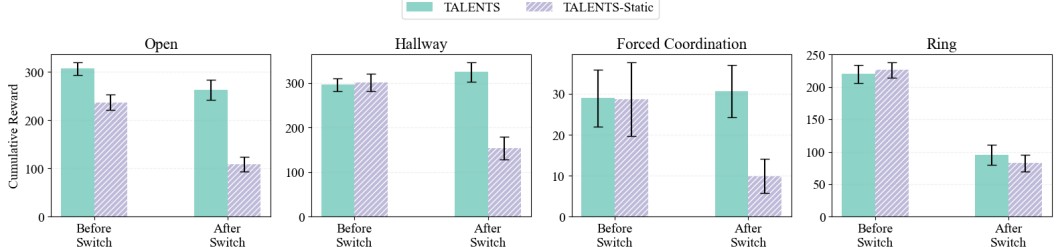

Figure 4: Accumulated reward with a partner policy swap midway through the episode ($t = 1200$). Error bars are one standard error from the mean.

Prolific and Cloud Research. Each participant completed three rounds of gameplay, each lasting for four minutes, with different agents as the partner. The entire session took around 30 minutes to complete and participants received a $7 base payment plus up to a $3 bonus depending on their performance. For these experiments, we limited the action frequency of agents to match that of human capability. Team score and subjective ratings were recorded as the performance metrics.

### 5.2.1 Experiment Design

A mixed experiment design is used for the human-agent evaluation, where agent type is the within subject variable (three levels: TALENTS, **GAMMA**, **BR**) and the map layout is the between subject variable (three levels: hallway, open, forced coordination). At the end of each round, participants are asked to complete a set of questionnaires measuring their subjective preference for the agent partner they just interacted with. The questionnaire consists of five surveys, each measuring their workload (NASA-TLX), perceived team fluency, trust, coordination, and satisfaction on 7-point Likert scales.

### 5.2.2 Results

We collected data from a total of 119 participants. To control data quality, we filtered out team trajectories from inactive participants and survey responses from those who incorrectly answered the trap questions. As shown in Fig 5, our agent significantly outperforms both baselines in team score and subjective ratings. Mixed ANOVA tests show significant main effects for team score ($F(2, 166) = 5.76, p = .003$), with our agent achieving significantly higher score than both baselines in post-hoc comparisons ($p < .05$). Similarly, our agent receives higher subjective ratings in team fluency ($F(2, 122) = 4.31, p = .02$) and perceived trust ($F(2, 122) = 3.23, p = .04$) compared to than the BR baseline ($ps < .05$).

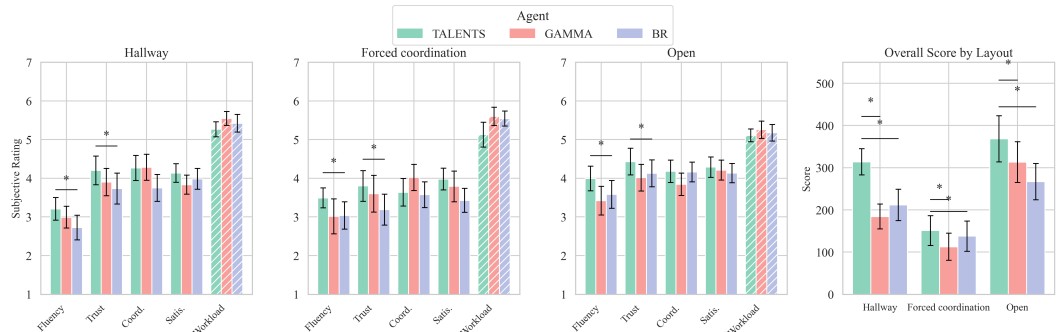

Figure 5: Human-agent teamwork evaluation comprises team scores and participants' subjective ratings of agent teammates. Perceived workload (shaded) is lower if better. Statistically significant differences between agents are marked by asterisks. Error bars are one standard error from the mean.

# 6  Conclusion

In this paper, we proposed TALENTS, a method for training a strategy-conditioned intra-episodic adaptive cooperator via generative teammates. Our approach combines a variational autoencoder to learn a latent strategy space, unsupervised clustering to identify distinct strategy types, and a fixed-share algorithm for online adaptation. The experimental results in the Overcooked environment demonstrate that our strategy-conditioned cooperator effectively identifies and adapts to partner strategies, achieving better performance than naive cooperators and significantly outperforming self-play agents trained with different partners. Our approach addresses a key limitation of existing methods by enabling intra-episodic adaptation to previously unseen and continuously changing partner strategies at test time. By representing strategies in a continuous latent space and clustering them into discrete types, we create a best-response mapping that allows for flexible adaptation. The fixed-share algorithm's ability to account for non-stationary partner behavior further enhances our method's robustness in dynamic collaborative settings.

## 6.1  Limitations

Our approach shows promising performance in the Overcooked environment but has limitations. It struggles on the Forced Coordination layout due to sparse reward signals when paired with ineffective or low-skill partners, which additionally causes the priority sampling method to bias toward choosing those partners in future episodes. Generalization to novel teammates is limited by the VAE's interpolation capabilities, meaning our agent can only truly generalize to new teammates if they exhibit similar behaviors to the training data. However, we demonstrate that our agent can successfully play with humans without having been trained on real human data. Furthermore, to better align with human gameplay, which requires real-time interactions, we artificially slowed down agent actions to match human actions-per-second rates. While agents act at every step, with the game designed for ten steps per second, humans typically only take 3–4 actions in the same timeframe. This slowdown may hinder agents' long-term planning capabilities.

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

# A Human Experiment Details

## A.1 Experimental Design

Human study participants were initially acquainted with the premise of the domain via a series of instruction screens (Figs A.1, A.2, A.3), then had the opportunity to test the game mechanics through a single-player tutorial session (Fig A.4a). In the gameplay phase of the experiment (Fig A.4b), participants were randomly assigned to one of the three human-study layouts (open, hallway, forced coordination). Each participant then played three games, one with each evaluated agent (TALENTS, GAMMA, and Population Best Response), presented in a randomized order to control for sequence effects. Episode length was $T = 2400$ environment timesteps (4 minutes). Many previous works in the Overcooked domain use a game length of $T = 400$ or $T = 600$ [34; 22; 39; 2]. We selected a longer game-length both due to the longer-horizon tasks of the modified Overcooked environment as well as to better compare the intra-episodic team adaptation capability of the algorithms and reduce the amount of randomness in our results.

Upon completion of each game, participants were asked to complete a post-game survey, consisting of NASA-TLX, team fluency, team trust, coordination, and satisfaction surveys, shown in Figure A.5.

## A.2 Online Data Collection Instructions and Interface

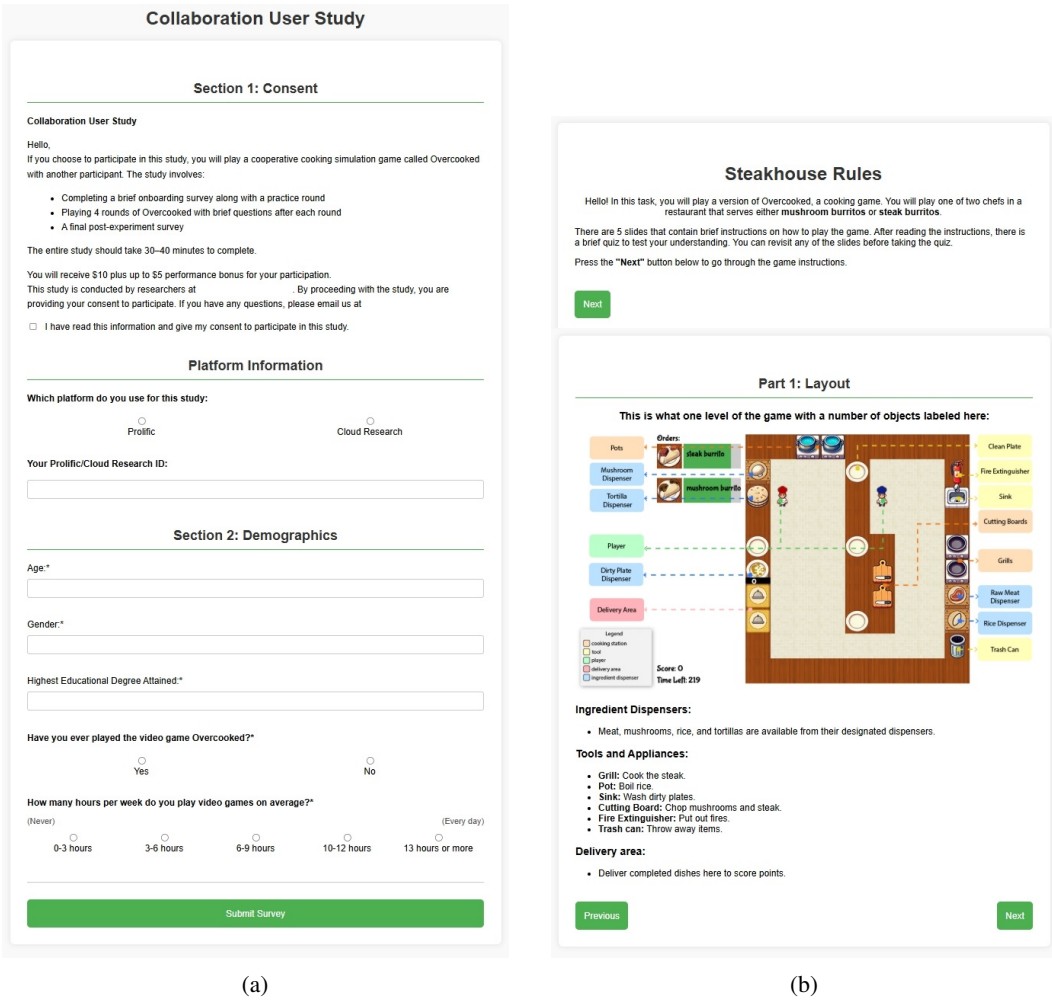

(a)             (b)

Figure A.1: Online data collection instructions.

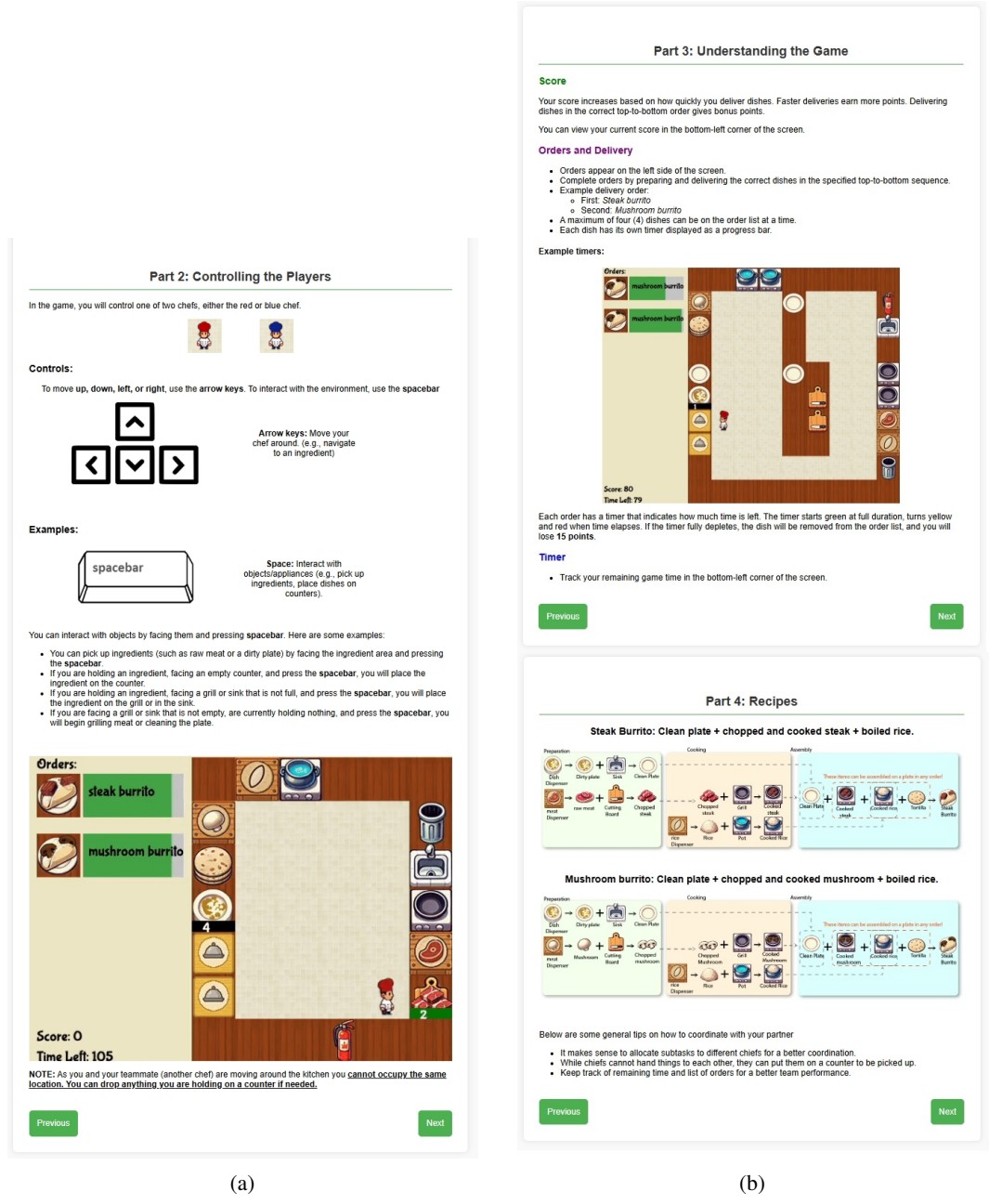

(a)                                                                                          (b)

Figure A.2: Online data collection instructions.

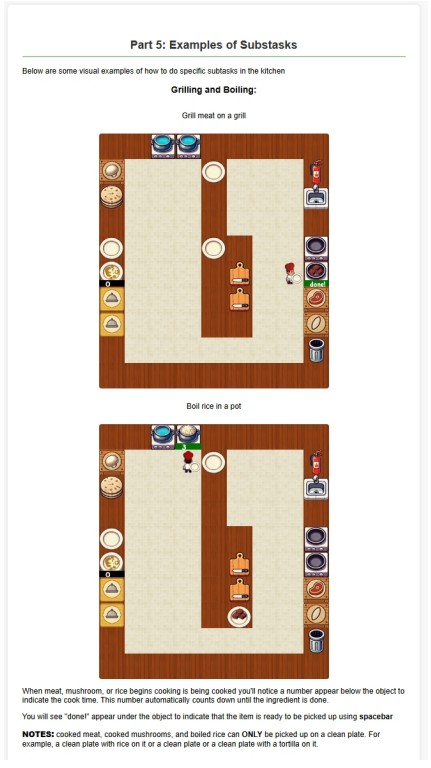

Figure A.3: Online data collection instructions.

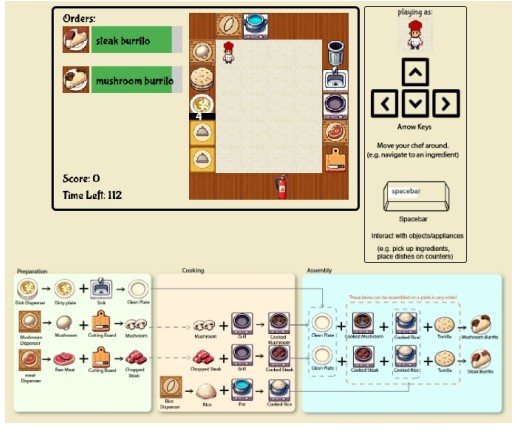

(a) Single-player tutorial session.

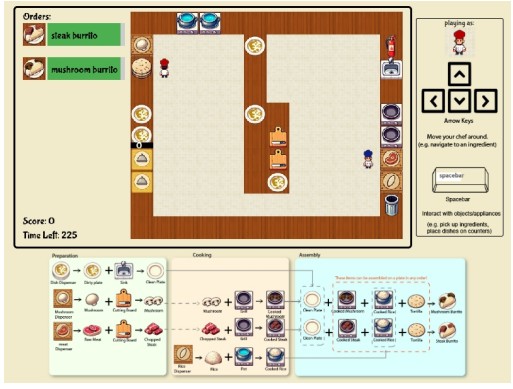

(b) Two-player (agent in blue, human in red) actual data collection session (hallway layout).

Figure A.4: Online data collection gameplay interfaces.

## Post-game Survey

### Section 1: NASA-TLX

**1. How mentally demanding was the task?**

(Very low)     (Very high)

1   2   3   4   5   6   7

**2. How hurried or rushed was the pace of the task?**

(Very low)     (Very high)

1   2   3   4   5   6   7

**3. How successful were you in accomplishing what you were asked to do?**

(Very low)     (Very high)

1   2   3   4   5   6   7

**4. How hard did you have to work to accomplish your level of performance?**

(Very low)     (Very high)

1   2   3   4   5   6   7

**5. How insecure, discouraged, irritated, stressed, and annoyed were you?**

(Very low)     (Very high)

1   2   3   4   5   6   7

### Section 2: Team Fluency

**1. The team worked fluently together.**

(Strongly disagree)     (Strongly agree)

1   2   3   4   5   6   7

**2. The collaboration contributed to the fluency and better performance of the interaction.**

(Strongly disagree)     (Strongly agree)

1   2   3   4   5   6   7

**3. The team's fluency improved over time.**

(Strongly disagree)     (Strongly agree)

1   2   3   4   5   6   7

**4. The interaction felt natural and effortless.**

(Strongly disagree)     (Strongly agree)

1   2   3   4   5   6   7

**5. You felt synchronized with your teammate's actions.**

(Strongly disagree)     (Strongly agree)

1   2   3   4   5   6   7

(a)

### Section 3: Team Trust

**1. My teammate was trustworthy.**

(Strongly disagree)     (Strongly agree)

1   2   3   4   5   6   7

**2. My teammate's actions were reliable and predictable.**

(Strongly disagree)     (Strongly agree)

1   2   3   4   5   6   7

**3. My teammate was committed to the task.**

(Strongly disagree)     (Strongly agree)

1   2   3   4   5   6   7

**4. I felt confident in my teammate's abilities.**

(Strongly disagree)     (Strongly agree)

1   2   3   4   5   6   7

**5. Did you feel comfortable depending on your teammate?**

(Strongly disagree)     (Strongly agree)

1   2   3   4   5   6   7

### Section 4: Co-ordination

**1. My teammate contributed equally to the team's performance.**

(Strongly disagree)     (Strongly agree)

1   2   3   4   5   6   7

**2. I had to carry the weight to make the team better.**

(Strongly disagree)     (Strongly agree)

1   2   3   4   5   6   7

**3. I was the most important member of the team.**

(Strongly disagree)     (Strongly agree)

1   2   3   4   5   6   7

**4. I had to frequently adjust my actions to account for my teammate.**

(Strongly disagree)     (Strongly agree)

1   2   3   4   5   6   7

(b)

### Section 5: Satisfaction

**1. I enjoyed the gameplay experience.**

(Strongly disagree)     (Strongly agree)

1   2   3   4   5   6   7

**2. I felt frustrated at times during the collaboration.**

(Strongly disagree)     (Strongly agree)

1   2   3   4   5   6   7

**3. The task was engaging and immersive.**

(Strongly disagree)     (Strongly agree)

1   2   3   4   5   6   7

**4. I felt satisfied with the overall team performance.**

(Strongly disagree)     (Strongly agree)

1   2   3   4   5   6   7

Submit Survey

(c)

Figure A.5: Post-game survey questions.

# B  Training Details

## B.1  Population Training

For the Behavior Preference [44] agent population, we define a set of nine tunable preferences, separated into three distinct categories. Each preference has the option of an "encouraged" reward shaping term as well as a "discouraged" term. For each category, we train agents with every permutation of the preferences (8 agents per category, 24 agents total). A best response agent is trained in self-play with each of the behavior-preferenced agents. We use these agents as the base population in human experiments following Wang et al. [44]'s observation that these agents cover a more diverse strategy space than other population methods (MEP, FCP).

| Preference | Reward Shape |
|---|---|
| Plating Ingredients | $-30, 20$ |
| Washing Plates | $-30, 20$ |
| Delivering Dishes | $-30, 20$ |
| Chopping Ingredients | $-30, 20$ |
| Potting Rice | $-30, 20$ |
| Grilling Meat/Mushroom | $-30, 20$ |
| Taking Mushroom From Dispenser | $-15, 10$ |
| Taking Rice From Dispenser | $-15, 10$ |
| Taking Meat From Dispenser | $-15, 10$ |

Table A.1: Event-based BP features and corresponding reward design.

For MEP [51] and FCP [39] agent populations, we train eight unique agents, saving two intermediate policy checkpoints ($\frac{1}{3}, \frac{2}{3}$ of total training iterations) and the final policy to form a total population of 24 policies.

## B.2  Encoder Training

As mentioned in the main text, we pair agents in the population and perform several joint rollouts to form the trajectory dataset. For FCP and MEP populations, we collect trajectories from every combination of policies in the population. For the BP population, we play each preference agent with its self-play best response. In total, each dataset of trajectories contained between 500 and 700 agent games with 2400 timesteps each. Encoder training hyperparameters are listed in table A.2, and were tuned via a simple linesearch. Encoder input observations were in the form of a 26-channel egocentric lossless state representation, one of the standard options of the Overcooked domain. KL weight $\beta$ was linearly annealed over the course of training.

## B.3  Cooperator Training

Cooperator agents were trained via Independent PPO, though other RL methods would likely also work. Action masking was applied during both train and test time to ensure agents chose valid actions. To encode the cluster information in the TALENTS agent, an action biasing approach was used. We experimented with using an embedding layer to encode the cluster parameters and appending that to the agent's observation, but found that approach to be less effective at learning distinguishably different behaviors for different types of partners, and as a result, had worse performance. Training hyperparameters can be found in table A.3. To test the belief update and expert switching mechanism of fixed share, we tracked the distribution of weights over time when switching the cooperator's partner's policy for one with a different strategy during an episode (same premise as the ablation study shown in 4). As shown in A.6, we see the best predicted expert change over time.

Table A.2: Role Encoder Training Parameters and Model Configuration

| Parameter | Default Value | Description |
|---|---|---|
| *Model Architecture* | | |
| Latent Dimension | 8 | Dimension of the latent role space |
| Window Length | 50 | Length of trajectory input sequence |
| Prediction Horizon | 50 | Future action prediction steps |
| *CNN Feature Extractor* | | |
| Input Channels | 26 | Channels in state representation |
| Conv Layers | [16, 32, 32] | Convolutional layer filter counts |
| Kernel Size | $3 \times 3$ | Convolution kernel dimensions |
| Activation | ReLU | Convolutional layer activation |
| *Sequence Encoding* | | |
| Action Embedding Dim | 8 | Discrete action embedding size |
| Temporal Encoder | GRU | Sequence processing architecture |
| GRU Hidden Size | 256 | Temporal encoder hidden dimension |
| GRU Input Size | CNN + Action | Combined feature input |
| *Encoder Network* | | |
| Hidden Layer Size | 256 | Encoder hidden dimension |
| Output Layer | $2\times$ Latent Dim | Mean and log-variance outputs |
| Encoder Activation | ReLU | Hidden layer activation |
| *Decoder Network* | | |
| Decoder RNN | GRU | Future sequence decoder |
| Decoder Hidden Size | 256 | Decoder RNN hidden dimension |
| Decoder Input | Latent + CNN | Combined latent and observation features |
| Action Predictor | Linear | Final action classification layer |
| *Training Hyperparameters* | | |
| Batch Size | 512 | Training batch size |
| Number of Epochs | 100 | Total training epochs |
| Learning Rate | $5 \times 10^{-4}$ | Adam optimizer learning rate |
| $\beta$ Start | 0.0 | Initial $\beta$-VAE regularization weight |
| $\beta$ End | 0.05 | Final $\beta$-VAE regularization weight |
| Train/Validation Split | 80/20 | Data split ratio (%) |

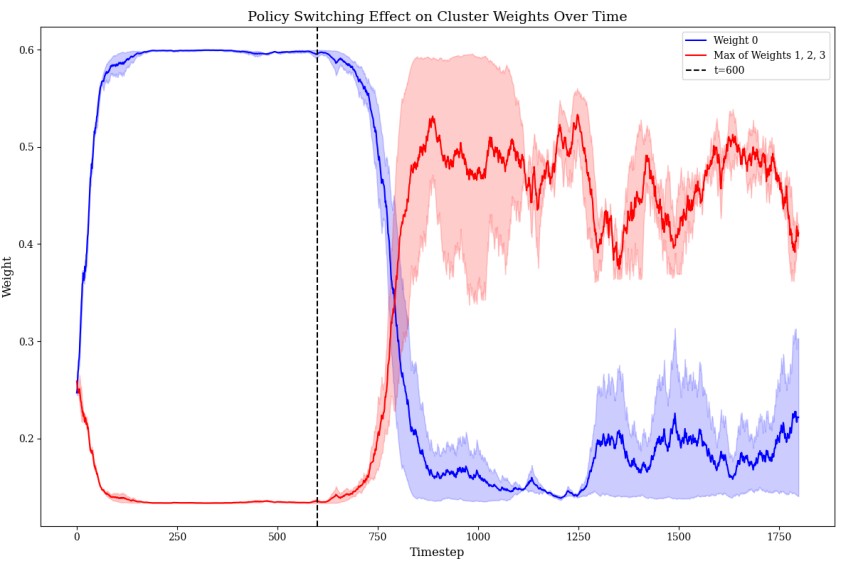

Figure A.6: Cluster Weight Distribution With a Partner Policy Swap Over Time

Table A.3: Cooperator Policy Training Parameters

| Parameter | Value |
|---|---|
| **Optimization Parameters** | |
| Total Training Steps | 4e7 |
| Training Batch Size | 38,400 |
| Parallel Environments | 16 |
| Learning Rate Schedule | $[10^{-3}, 10^{-4}, 10^{-5}]$ |
| GAE Lambda ($\lambda$) | 0.99 |
| KL Coefficient | 0.5 |
| PPO Clip Parameter | 0.2 |
| Value Function Loss Coefficient | 0.5 |
| Entropy Coefficient | 0.02 |
| Gradient Clipping | 1.0 |
| Discount Factor ($\gamma$) | 0.995 |
| **Fixed Share Parameters** | |
| Learning Rate | 0.2 |
| Sharing Parameter ($\alpha$) | 0.4 |
| Regret Clipping | 1.0 |
| **Network Architecture** | |
| Actor Embedding Dimension | 392 |
| Critic Embedding Dimension | 392 |
| Cluster Embedding Dimension | 32 |
| Action Space Size | 27 |
| Observation Channels | 12 |
| Action Bias Weight | 2.0 |

# C   Additional Experiments

## C.1   Additional Baselines

The primary motivation of this work was to develop a state-of-the-art method for training a cooperator using simulated data generated by agent populations. As such, the baseline algorithms used for evaluation (GAMMA, BR) were selected as they are representative SOTA methods for population-based training, and enable us to isolate the impact of our novel training and adaptation methodology. However, it is still valuable to compare against other leading zero-shot coordination methods, notably ones which also utilize strategy inference or partner modeling modules.

We evaluate against E3T, a self-play method that uses a partner modeling module to anticipate its teammate's future intentions, and PACE, an adaptation framework that injects a peer identification reward term to encourage agents to choose actions which allow them to form more accurate estimates about their teammates' types. We report the results of preliminary experiments in A.4.

We find that TALENTS remains the top performer out of all the assessed methods, while E3T is able to outperform the remaining baselines. PACE underperforms, achieving comparable scores to the population best response baseline. This can likely be attributed to the fact that PACE is an effective *inter*-episodic adaptation framework. However, in our zero-shot coordination problem setting, PACE is unable to form its belief in the short time frame necessary for high performance, and therefore achieves a similar score as the naive population best response baseline. In contrast, TALENTS is able to perform *intra*-episodic adaptation due to its tracking regret minimization module.

| Algorithm | Score (± Std. Err.) |
|-----------|---------------------|
| E3T | 548.66 ± 25.44 |
| PACE | 313.34 ± 18.48 |
| GAMMA | 374.10 ± 24.99 |
| BR | 343.71 ± 24.27 |
| TALENTS | 605.88 ± 27.89 |

Table A.4: Average performance over the evaluated layouts, comparing the additional baselines of E3T and PACE.

## C.2 Effect of Cluster Count

We use silhouette analysis in order to partition the VAE latent space into discrete strategy clusters in an unsupervised manner. However, there is no theoretical guarantee that the optimal cluster count as determined by silhouette analysis will result in optimal performance when those clusters are used to train a TALENTS cooperator. To empirically evaluate our approach, we conduct a sensitivity analysis over cluster count, training and evaluating cooperator agents using various numbers of latent clusters. The results of the sensitivity analysis are shown in A.5, while the corresponding silhouette scores are shown in A.7. We find that the silhouette-optimal number of clusters achieves the highest score, suggesting that silhouette scores are an appropriate metric to use for this application.

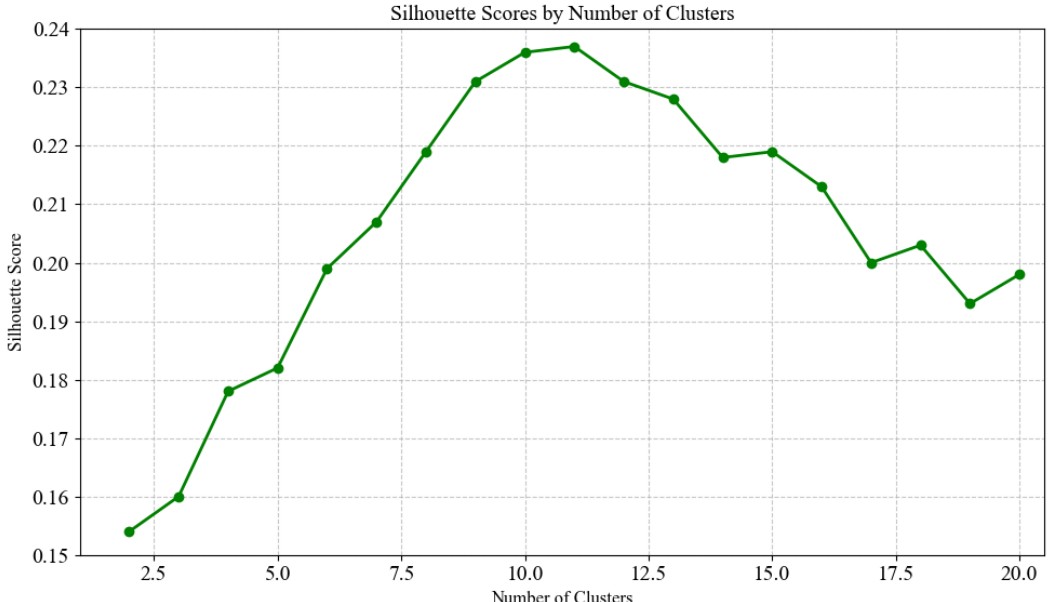

Figure A.7: Silhouette analysis of a VAE latent space in the open layout. Higher silhouette score indicates more optimal clustering, with 11 being the optimal number for this latent space.

| Number of Clusters | Score (± Std. Err.) |
|--------------------|---------------------|
| 1 | 650.83 ± 28.00 |
| 2 | 701.68 ± 11.93 |
| 4 | 620.35 ± 39.01 |
| 8 | 740.93 ± 79.15 |
| 11 | 865.82 ± 15.11 |
| 14 | 724.94 ± 40.60 |
| 16 | 648.57 ± 26.26 |

Table A.5: Effect of different numbers of clusters on overall performance of a TALENTS cooperator, evaluated in the open layout.

