# OpenReview forum: "Adaptively Coordinating with Novel Partners via Learned Latent Strategies"
_NeurIPS.cc/2025/Conference — NeurIPS 2025 poster_

### Official Review · Reviewer_LHkq · 2025-06-29

**Clarity:** 3
**Significance:** 2
**Originality:** 2
**Rating:** 4
**Confidence:** 5

**Summary:**

This paper tackles the challenge of adaptation in human-agent collaboration, where agents must adjust to diverse and dynamically changing partner strategies. The authors propose a strategy-conditioned cooperator framework that uses a variational autoencoder to learn a latent strategy space from partner trajectories. Strategies are categorized via clustering, and the agent is trained to cooperate with different strategy types. During interaction with a novel partner, the agent adapts online using a fixed-share regret minimization algorithm that continuously updates its estimate of the partner’s strategy. The method is evaluated in a modified Overcooked environment and a human user study, showing improved performance over existing baselines when cooperating with both artificial and human partners.

**Questions:**

1. Could the authors provide a more comprehensive discussion of the related work, particularly the relevant studies mentioned above (e.g., E3T, PACE, GSCU, ProAgent)? In addition, an empirical comparison with these closely related baselines, especially those also evaluated in the Overcooked domain, would significantly clarify the contribution and positioning of this work.

2. Could the authors consider extending their framework to competitive settings, as explored in works like PACE and GSCU? The current structure, in which the ego agent infers the partner's behavior and computes a best response, appears well-suited for adaptation in competitive scenarios, such as poker or adversarial games. Furthermore, demonstrating the method's effectiveness in more complex environments with image-based observations and higher-dimensional action spaces would considerably strengthen the paper.

3. In Figure 4, TALENTS shows consistently higher scores in fluency and trust, but does not significantly outperform baselines in coordination and satisfaction. Could the authors elaborate on the reasons behind this discrepancy?

**Ethical Concerns:**

["NO or VERY MINOR ethics concerns only"]

**Final Justification:**

The additional discussions and baseline results help address the initial lack of a thorough literature review and the unclear contribution. However, the extension to competitive settings remains an open direction for future work. A rating of 4: Borderline Accept reflects the current quality of the paper.

**Limitations:**

See Weaknesses and Questions.

**Paper Formatting Concerns:**

No formatting concerns.

**Quality:**

2

**Strengths And Weaknesses:**

**Strengths**
1. The paper is well-written and easy to follow. The figures and pseudocode are generally helpful in conveying the methodology and implementation details.

2. The problem of real-time adaptation in human-AI collaboration is important and well-motivated. The proposed approach, which enables an agent to infer its partner’s strategy and adapt its policy accordingly, addresses a key challenge in this domain. The integration of the fixed-share regret minimization algorithm within TALENTS to reduce regret is both principled and effective.

3. The inclusion of experiments with human participants strengthens the paper by demonstrating the practical utility of the method in real-world human-ai collaborative scenarios.


**Weaknesses**
1. The paper lacks a thorough discussion of related work. Teammate modeling for zero-shot coordination and real-time adaptation has been extensively studied. For example, E3T [1] introduces an end-to-end approach with a partner modeling module to predict the partner’s actions based on historical context. PACE [2] explicitly addresses both teammate and opponent adaptation and similarly encodes behavioral context to update the ego agent’s policy. GSCU [3] also tackles adaptation via a CVAE-based latent policy encoder and online Bayesian updates, using an EXP3 algorithm to manage regret—an approach conceptually close to this work. More recently, ProAgent [4] incorporates LLMs with belief inference for zero-shot coordination. Notably, PACE and ProAgent both evaluate their methods in the Overcooked domain, making them particularly relevant baselines. The omission of these works from the related work and experimental comparisons limits the clarity of this paper’s contribution and novelty.

2. The empirical evaluation is limited to a single environment (Overcooked), which is a grid-based domain with relatively low-dimensional state and action spaces. Additional experiments in more complex or diverse collaborative environments would help demonstrate the generality of the approach.

3. The proposed method is evaluated only in cooperative settings. In contrast, related works such as PACE and GSCU consider both cooperative and competitive dynamics. Addressing broader multi-agent interactions would enhance the applicability of the framework.

[1] Yan, Xue, et al. "An efficient end-to-end training approach for zero-shot human-AI coordination." Advances in Neural Information Processing Systems 36 (2023): 2636-2658.

[2] Ma, Long, et al. "Fast peer adaptation with context-aware exploration." arXiv preprint arXiv:2402.02468 (2024).

[3] Fu, Haobo, et al. "Greedy when sure and conservative when uncertain about the opponents." International Conference on Machine Learning. PMLR, 2022.

[4] Zhang, Ceyao, et al. "Proagent: building proactive cooperative agents with large language models." Proceedings of the AAAI Conference on Artificial Intelligence. Vol. 38. No. 16. 2024.

---

> ### Author Rebuttal · Authors · 2025-07-31
>
> We thank the reviewer for their comprehensive feedback of our work. We greatly appreciate the reviewer for highlighting some of the strengths of our work, including that our method is __well-motivated, effective, and grounded in principled approaches__. We are encouraged that our human-agent user study lends credibility to the real-world applicability of the research, and that the paper is well-written and easy to follow. Below, we address the weaknesses and questions raised by the reviewer.
>
> ## Discussion on Related Work
> We greatly appreciate the reviewer’s suggestions with respect to the discussion of related work. We have expanded our discussion, particularly in the Strategy Inference section to highlight E3T, the bayesian adaptation framework of GSCU, the contextual encoder and exploration reward of PACE, as well as the LLM-based belief inference method of ProAgent. We hope that this additional discussion will provide a clearer understanding of the body of work that has informed and contributed to the development of TALENTS. In addition, we hope that this discussion will provide a clearer understanding as to the novelty and contributions of our method.
>
> Furthermore, in addition to discussing them in the related work section, __we have prepared E3T and PACE as additional baselines in our experiment section__ (see next response for details)
>
> ## Additional Baselines
> In relation to the previous comment, we recognize that evaluating our method against the aforementioned methods is important to provide better context and positioning of our novel TALENTS agent. Based on the reviewer’s recommendation, we have decided to implement __E3T__ and __PACE__ as additional baselines. We chose not to evaluate GSCU, as it was included in the PACE paper as a baseline and was significantly outperformed by PACE in the Overcooked setting [1]. The following are our preliminary results from the new baselines:
>
> | Model   | Score (± Std. Err.)     |
> |---------|-------------------------|
> | E3T     | 548.66 ± 25.44          |
> | PACE    | 313.34 ± 18.48          |
> | GAMMA   | 374.10 ± 24.99          |
> | BR      | 343.71 ± 24.27          |
> | TALENTS | 605.88 ± 27.89          |
>
> __TALENTS remains the top performer out of all the assessed methods__, while E3T is able to outperform all the other baselines. PACE underperforms, achieving comparable scores to the population best response baseline. PACE, similar to other bayesian adaptation methods like GSCU and MeLIBA [2], are effective in the __inter__-episodic setting, where the agent is able to form beliefs over __several episodes of play with the same agent__, transitioning from exploratory to exploitative behaviors over time. In our zero-shot coordination setting, in which agents must immediately adapt to unseen partners, PACE is unable to form its belief in the short time frame necessary for high-performance, and therefore achieves a similar score as the naive population best response baseline. This highlights a key differentiator and source of novelty of TALENTS, which is its ability to perform __intra__-episodic adaptation. __This capability is largely made possible by the fixed-share tracking regret minimization method__ (Algorithm 2), and is a reason why TALENTS achieves SOTA performance as a zero-shot coordinator.
>
> ## Extension to Competitive Settings
> We appreciate the reviewer’s suggestion to explore competitive or adversarial settings. While we agree these are valuable directions, our work is focused on the __zero-shot coordination / ad-hoc teaming problem in cooperative environments__, where the goal is rapid adaptation to unfamiliar partners. Introducing competition would shift the scope to a different class of problems involving adversarial intent and asymmetric payoffs, requiring distinct modeling and evaluation. That said, we agree that extending TALENTS to mixed-motive or adversarial domains is a promising avenue for future research, and __we now note this in the conclusion__.
> We also agree with the reviewer’s point on environment complexity. This was a key motivation for utilizing a modified version of Overcooked, as __the original setting [4] only had limited opportunities for behavioral diversity__. While recent extensions [3, 1] introduced more difficulty, __our modified environment introduces considerable complexity__ through the inclusion of larger maps, longer episodes (T=240s rather than the typical T=40s [9,4] or T=60s [3,8]), temporal constraints, and multi-step tasks, enabling a more intricate evaluation of agent robustness in highly complex teamwork tasks.
> While using image-based observations is feasible, we chose the grid-based representation of Overcooked to enable a smaller model footprint, requiring fewer parameters and less data. Moreover, the structured representation focuses the learning signal, making it easier to analyze and ablate the key factors that contribute to our method’s success. That said, we agree this raises an interesting direction for future work and would be more akin to real-world applications.
>
> ## User Study Survey Scores
>
> This is a great question, we thank the reviewer for raising this discrepancy. The survey responses in the trust and fluency category are typically indicative of the human’s perception of their agent partner’s capability and proficiency [6,7], whereas the coordination and satisfaction metrics tend to be more indicative of the human’s perception of the joint teamwork [5]. We hypothesize that the lack of statistically significant ratings in these regards may be due to the __distribution shift between agent-agent games and human-agent games__. Agents tend to move differently in the Overcooked environment compared to human players, and may perform actions perceived as unpredictable or erratic to their human teammates. Since our cooperator agents were only paired with other agents during training, __there was no mechanism to incentivize them to behave in a “human-interpretable” way__. __This shift is equally present among the three methods assessed in the user study__. To further assess this hypothesis, we would need to ask more post-game questions (particularly open-ended questions) to participants, which was beyond the scope of the initial human subject study.
>
> [1] Ma, L., Wang, Y., Zhong, F., Zhu, S. C., & Wang, Y. (2024). Fast peer adaptation with context-aware exploration. arXiv preprint arXiv:2402.02468.
>
> [2] Zintgraf, L., Devlin, S., Ciosek, K., Whiteson, S., & Hofmann, K. (2021). Deep interactive bayesian reinforcement learning via meta-learning. arXiv preprint arXiv:2101.03864.
>
> [3] Liang, Y., Chen, D., Gupta, A., Du, S. S., & Jaques, N. (2024). Learning to cooperate with humans using generative agents. Advances in Neural Information Processing Systems, 37, 60061-60087.
>
> [4] Carroll, M., Shah, R., Ho, M. K., Griffiths, T., Seshia, S., Abbeel, P., & Dragan, A. (2019). On the utility of learning about humans for human-ai coordination. Advances in neural information processing systems, 32.
>
> [5] Richard M Ryan and Edward L Deci. Self-determination theory and the facilitation of intrinsic motivation, social development, and well-being. American psychologist, 55(1):68, 2000
>
> [6] Guy Hoffman. Evaluating fluency in human–robot collaboration. IEEE Transactions on Human-Machine Systems, 49(3):209–218, 2019
>
> [7] Richard M Ryan and Edward L Deci. Self-determination theory and the facilitation of intrinsic motivation, social development, and well-being. American psychologist, 55(1):68, 2000
>
> [8] Strouse, D. J., McKee, K., Botvinick, M., Hughes, E., & Everett, R. (2021). Collaborating with humans without human data. Advances in neural information processing systems, 34, 14502-14515.
>
> [9] Zhao, R., Song, J., Yuan, Y., Hu, H., Gao, Y., Wu, Y., ... & Yang, W. (2023, June). Maximum entropy population-based training for zero-shot human-ai coordination. In Proceedings of the AAAI Conference on Artificial Intelligence (Vol. 37, No. 5, pp. 6145-6153).

---

> > ### Comment · Reviewer_LHkq · 2025-08-05
> >
> > Thanks for the detailed response. Most of my concerns have been addressed, and I will raise my rating accordingly. I encourage the authors to incorporate the additional baselines and discussions into the paper to further enhance its quality and clarity.

---

> > > ### Author Response · Authors · 2025-08-07
> > >
> > > Dear Reviewer LHkq,
> > >
> > > Thank you for your insightful feedback, we appreciate your expertise in this area and time taken to review our submission. We have updated our manuscript with the additional baselines and discussions on related methods, survey scores, and competitive settings. If you have any remaining questions or avenues for improvement, we would be happy to discuss further.
> > >
> > > Best regards,
> > > The Authors

---

### Official Review · Reviewer_xno3 · 2025-07-01

**Clarity:** 3
**Significance:** 3
**Originality:** 2
**Rating:** 4
**Confidence:** 2

**Summary:**

This paper introduces TALENTS, a novel framework for zero-shot human-agent and agent-agent coordination based on learned latent strategy embeddings. It employs a variational autoencoder to model teammate strategies, clusters these strategies in latent space, and trains a cooperator agent conditioned on the resulting clusters. Online adaptation is achieved through a fixed-share regret minimization algorithm. The method is evaluated on an extended version of the Overcooked environment and tested with both synthetic agents and human subjects.

**Questions:**

1. Including results from at least one additional environment—even preliminary—would strengthen the paper’s claim of broad relevance in human-AI coordination.

2. Performing a sensitivity analysis over the key hyperparameters such as the number of the clusters would enhance confidence in the robustness and reliability of this component.

3. Including an ablation where the system encounters humans with atypical strategies would demonstrate robustness and validate assumptions about the latent space.

4. Adding more Ablation Studies and comparisons with alternative approaches would situate the proposed method more clearly in the literature.

**Ethical Concerns:**

["NO or VERY MINOR ethics concerns only"]

**Final Justification:**

Thanks for the responses! I maintain my score.

**Limitations:**

yes

**Quality:**

2

**Strengths And Weaknesses:**

**Strengths:**

1. **Novelty and Motivation**: The approach of clustering latent behaviors and adapting via fixed-share regret minimization is original and well-motivated within the context of ad hoc teamwork and zero-shot coordination.

2. **Technical Soundness**: The combination of VAE-based behavior modeling with a regret-minimization framework is technically coherent. The fixed-share approach to handle intra-episodic partner behavior shifts is well-justified.

3. **Empirical Results**: TALENTS achieves strong performance across most Overcooked layouts, particularly outperforming GAMMA and best-response agents in most settings. The human-agent evaluation demonstrates promising real-world applicability, with higher subjective ratings on fluency, trust, and satisfaction.

4. **Inclusion of Human-Agent Study**: The paper includes an online user study with human participants, which is crucial for validating coordination methods in realistic settings. This adds significant value and addresses a critical aspect of human-AI interaction.

---

**Weaknesses and Points for Improvement:**

1. **Limited Generalization Beyond Overcooked**: All experiments are confined to a single domain - Overcooked. While this is a standard testbed, it is unclear whether the method would generalize to other complex multi-agent settings. A discussion of this limitation or preliminary results on a second environment would improve the work’s impact.

2. **Clustering Sensitivity**: Details on Strategy Clustering: While K-Means with silhouette analysis is used to determine optimal clusters, there is no guarantee that the identified clusters correspond to meaningful or semantically coherent behavioral categories. The sensitivity of the overall system's performance to the chosen number of clusters (K) and whether these discrete clusters adequately represent the continuous spectrum of human strategies could be further investigated. The implications of misclassifying a human's strategy are not explicitly discussed.

3. **Latent Space Generalization**: While the VAE is trained on diverse agent behaviors, its ability to encode novel human-like strategies not seen during training is questionable. The paper assumes that the VAE latent space covers the full spectrum of possible strategies, but this is not empirically validated. This assumption may fail in real-world cases (e.g., humans with unique strategies), and the paper does not quantify how performance degrades in such cases, especially from humans who may behave in fundamentally different ways than the baseline agents.

4. **Fixed-Share Regret Minimization**: While the fixed-share algorithm allows for tracking non-stationary strategies, its effectiveness is only demonstrated in one synthetic experiment (partner policy swap mid-episode). It would be helpful to see more extensive evaluation of this component, especially in human-agent interactions where partner policies are inherently non-stationary.

5. **Ablation Studies and Comparisons**: While the paper includes an ablation of the fixed-share component, there is limited analysis of other key design choices: 1) What is the performance when conditioning directly on continuous latents instead of discrete clusters? 2) How does the system behave without priority-based sampling or latent-space biasing? 3) GAMMA is a strong baseline, but including some other advanced related works, or simpler latent-based adaptation methods would strengthen the comparative picture.

---

> ### Author Rebuttal · Authors · 2025-07-31
>
> We would like to thank the reviewer for their thoughtful comments and suggestions. We are encouraged to hear that our method, TALENTS, has been perceived as __novel, original, and well motivated__. Additionally, we are excited to hear that our empirical results, particularly TALENTS' performance across Overcooked layouts and the human-agent evaluation, were seen as __strong and indicative of real-world applicability__. We also appreciate the recognition of the __technical soundness of our approach__, especially the integration of VAE-based behavior modeling with fixed-share regret minimization. In the following, we will address the weaknesses and questions of the reviewer:
>
> ## Limited Evaluation on Overcooked
> Thank you for requesting further information about the evaluation domain. Overcooked is a challenging domain for AI agents [1,2,3], as well as humans, providing a rich testbed for agent-agent and human-agent teaming capabilities due to various factors, such as time pressure, multiple cooking stations, multi-step recipes, and a reward structure that encourages fast deliveries.  While we agree that evaluating TALENTS in additional domains such as Hanabi (which introduces partial observability) or cooperative pursuit-evasion (which adds adversarial dynamics) would be valuable, incorporating such experiments is beyond the scope of this submission. We consider these exciting directions for future work and plan to explore them in follow-up studies.
>
> ## Cluster Sensitivity
> We appreciate the request for further insights into the number of clusters as well as the sensitivity of TALENTS with respect to the number of chosen clusters. To clarify, we utilize silhouette analysis to cluster our latent embedding space with the goal of finding a suitable representation of discrete strategy types. There is precedence of this method as an effective way of partitioning a latent embedding space [7,8]. To the reviewer’s point, prior work in opponent modeling and latent encoding methods have explored modifications to the VAE objective in order to learn a clustered embedding space in a more end-to-end fashion [10,9,6], but these objectives often require privileged information such as the identity of the agent from which the trajectory originated from in order to function [10,9]. __Unlike these methods, TALENTS does not make such assumptions__. To evaluate the efficacy of silhouette analysis for selecting an optimal number of clusters, we perform a __sensitivity analysis over a number of cluster partitions__, ranging from the naive one-cluster case to 16 clusters. __We find that the silhouette analysis optimal number of clusters (11) achieves the highest score, empirically demonstrating the strength of our clustering approach.__
>
> | Number of Clusters   | Score (± Std. Err.)     |
> |---------|-------------------------|
> | 1     | 650.83 ± 28.00          |
> | 2    | 701.68 ± 11.93         |
> | 4   | 620.35 ± 39.01          |
> | 8      | 740.93 ± 79.15          |
> | 11 | 865.82 ± 15.11          |
> | 14 | 724.94 ± 40.60          |
> | 16 | 648.57 ± 26.26          |
>
> The issue of potentially misclassifying a partner’s behavior and selecting an inappropriate cluster is a valid concern. To this end, we assess TALENTS’s ability to correctly identify player behaviors. We have conducted additional experiments in which TALENTS is paired with all static experts, __correctly identifying the correct expert in 91.3% of evaluations__.
>
> ## Generalization and Unseed Human Data
> Thank you for requesting further insights on TALENTS’s ability to handle novel human behavior. It is indeed correct that one assumption of TALENTS is that the VAE’s latent space sufficiently covers novel human behavior. The latent space is trained via a dataset of 11 reward-shaped policies. Evaluation is done using a population of 10 diverse unseen policies for all experiments to ensure meaningful generalization of our agent. There is no guarantee that these evaluation policies are necessarily represented by any particular cluster in the latent space, but the rationale of using a regret minimization framework for partner inference is the guarantee that __out-of-distribution policies will not cause catastrophic performance degradation due to the framework's theoretical bounds__. Furthermore, we analyzed the VAE’s ability, which is only trained on our population agents, to encode the data from our human experiments and found that human data covers a similar distribution in our VAE in terms of cluster representation as the behavior preference agents used to train the model. Calculating the Jensen-Shannon divergence of the agent and human data distributions yields 0.1516, __indicating a high degree of similarity between the two__. This value suggests that the __agent behavior closely approximates human behavior, with only minor differences__.
>
> ## Fixed-Share Regret Minimization
> We appreciate the request for a more in-depth evaluation of non-stationary partners and agree that this is an important area of analysis. As shown in Figure A.6, TALENTS is capable of adapting to novel partners within just a few seconds. Additionally, we show that TALENTS successfully identifies the correct expert in 91.3% of evaluation episodes. Given that the VAE effectively captures the range of human behaviors observed in our human-agent experiments, and that our human subject study reports high ratings for fluency, trust, and overall performance relative to other baselines (see Figure 4), we conclude that TALENTS performs robustly in settings with non-stationary partners (such as humans).
>
> ## Additional Baselines
> We appreciate the reviewer for pointing out the need for additional baselines and are excited to report that we have conducted additional experiments with two more baselines, namely E3T [4] and PACE [5]. We have added these baselines in addition to our prior evaluation of TALENTS against several leading zero-shot coordination baselines (GAMMA, MEP, FCP, BP). E3T [4] utilizes a modified self-play objective alongside a partner modeling module to perform ZSC, and PACE [5] adapts its behavior via a learned context encoder. Preliminary results are as follows:
>
>
> | Model   | Score (± Std. Err.)     |
> |---------|-------------------------|
> | E3T     | 548.66 ± 25.44          |
> | PACE    | 313.34 ± 18.48          |
> | GAMMA   | 374.10 ± 24.99          |
> | BR      | 343.71 ± 24.27          |
> | TALENTS | 605.88 ± 27.89          |
>
> __With these results, TALENTS demonstrates that it can, consistently, outperform other baselines, providing a significant improvement over existing literature.__
>
> [1] Liang, Y., Chen, D., Gupta, A., Du, S. S., & Jaques, N. (2024). Learning to cooperate with humans using generative agents. Advances in Neural Information Processing Systems, 37, 60061-60087.
>
> [2] Zhao, R., Song, J., Yuan, Y., Hu, H., Gao, Y., Wu, Y., ... & Yang, W. (2023, June). Maximum entropy population-based training for zero-shot human-ai coordination. In Proceedings of the AAAI Conference on Artificial Intelligence (Vol. 37, No. 5, pp. 6145-6153).
>
> [3] Carroll, M., Shah, R., Ho, M. K., Griffiths, T., Seshia, S., Abbeel, P., & Dragan, A. (2019). On the utility of learning about humans for human-ai coordination. Advances in neural information processing systems, 32.
>
> [4] Yan, X., Guo, J., Lou, X., Wang, J., Zhang, H., & Du, Y. (2023). An efficient end-to-end training approach for zero-shot human-AI coordination. Advances in neural information processing systems, 36, 2636-2658.
>
> [5] Ma, L., Wang, Y., Zhong, F., Zhu, S. C., & Wang, Y. (2024). Fast peer adaptation with context-aware exploration. arXiv preprint arXiv:2402.02468.
>
> [6] Zintgraf, L., Devlin, S., Ciosek, K., Whiteson, S., & Hofmann, K. (2021). Deep interactive bayesian reinforcement learning via meta-learning. arXiv preprint arXiv:2101.03864.
>
> [7] Zhao, M., Simmons, R., & Admoni, H. (2022, October). Coordination with humans via strategy matching. In 2022 IEEE/RSJ International Conference on Intelligent Robots and Systems (IROS) (pp. 9116-9123). IEEE.
>
> [8] Pandya, R., Zhao, M., Liu, C., Simmons, R., & Admoni, H. (2024, May). Multi-agent strategy explanations for human-robot collaboration. In 2024 IEEE International Conference on Robotics and Automation (ICRA) (pp. 17351-17357). IEEE.
>
> [9] Papoudakis, G., & Albrecht, S. V. (2020). Variational autoencoders for opponent modeling in multi-agent systems. arXiv preprint arXiv:2001.10829.
>
> [10] Aditya Grover, Maruan Al-Shedivat, Jayesh K Gupta, Yura Burda, and Harrison Edwards. Learning policy representations in multiagent systems. International Conference on Machine learning, 2018a.

---

### Official Review · Reviewer_Eztg · 2025-07-02

**Clarity:** 3
**Significance:** 2
**Originality:** 2
**Rating:** 4
**Confidence:** 4

**Summary:**

# Summary
The paper proposes a novel method for training an adaptive cooperative agent that explicitly adapt online by adjusting its internal strategy clusters. The method first learns a VAE from offline joint trajectories data. Then apply K-mean clustering on the VAE's latent space. The cooperative agent is then trained by sampling "partners" from the VAE clusters. During test-time, the "Fixed-Share" is used by to infer the current cluster of the partner, which overall gives the adaptivity to the cooperative agent. The method provide good empirical results across Overcooked layouts.

**Questions:**

See weaknesses

**Ethical Concerns:**

["NO or VERY MINOR ethics concerns only"]

**Final Justification:**

I've increased my score to 4 given the current scope and impact of the paper. The evaluation environments are currently limited environments and marginal performance improvement.

**Limitations:**

yes

**Paper Formatting Concerns:**

-

**Quality:**

2

**Strengths And Weaknesses:**

# Strengths
- The paper proposes a sophisticated and effective algorithm for online adaptation of a cooperative agent.
- The method is well motivated.
- Good empirical results.


# Weaknesses
- A baseline where the cooperative agent training does not require a VAE, e.g., MeLIBA, is not included. The gap between VAE- and non VAE-based (or even learned vs explicit) strategy inference methods would give a good sense of improvement the proposed method provides.
- I find the analysis to be lacking. It is very interesting that the proposed algorithm optimizes the experts per timestep. However, There is no analysis how the experts provide specialization or how often the switch happens internally. This analysis would still be useful even if the partner is static throughout an episode.
- No experiment on the impact of the offline data. Having this experiment would provide a useful observation whether the proposed method would still operate given different quantities and qualities of the offline data.
- Another ablation that is not included is training an RNN policy with algorithm 1 without using algorithm 2 during test-time. This baseline would provide information on how algorithm 2 performs compared to a simple RNN policy.

## Minor comments
- Figure 1 caption is not complete
- Line 68: incomplete sentence

I'm willing to increase the score if all the Weaknesses are properly addressed.

---

> ### Author Rebuttal · Authors · 2025-07-31
>
> We thank the reviewer for their thorough and insightful review. We appreciate that the reviewer found the proposed TALENTS method to be well-motivated, sophisticated, and effective, as well as being well-supported by empirical results. Below, we will address the critiques and weaknesses:
>
> ## Additional Baselines (MeLIBA, non-VAE methods):
> We thank the reviewer for their suggestion for additional baselines and evaluating the impact of VAE and non-VAE based methods. We agree that this is an important to evaluation. To this end, we currently evaluate against GAMMA [1] (a SOTA method which __uses a VAE__ for partner generation), as well as BP [2], FCP [3], and MEP [4] (SOTA population-trained methods which __do not use a VAE__ at all). The requested MeLIBA baseline falls into the same category as the GAMMA agents, using a VAE as part of its methodology [5]. However, to further extend our experiments, we have implemented E3T [6] as an additional baseline (a SOTA __self-play__ method which __does not use a VAE__ but has a __partner modeling module__ for strategy inference), as well as PACE [7] (a SOTA __population-trained__ method which uses a __context encoder__ to explore and adapt to partners). Overall results, including the new baselines, are as follows, showing strong performance of TALENTS as compared to other baselines on the open and hallway layouts.
>
> | Model   | Score (± Std. Err.)     |
> |---------|-------------------------|
> | E3T     | 548.66 ± 25.44          |
> | PACE    | 313.34 ± 18.48          |
> | GAMMA   | 374.10 ± 24.99          |
> | BR      | 343.71 ± 24.27          |
> | TALENTS | 605.88 ± 27.89          |
>
> With the inclusion of new baselines, TALENTS still maintains the highest performance out of all assessed methods, while E3T is able to outperform all the other baselines. PACE underperforms, achieving comparable scores to the population best response baseline. PACE, similar to other bayesian adaptation methods like GSCU [8] and MeLIBA [4], are effective in the __inter__-episodic setting, where the agent is able to form beliefs over __several episodes of play__ with the same agent, transitioning from exploratory to exploitative behaviors over time. In our zero-shot coordination setting, in which agents must immediately adapt to unseen partners, __PACE is unable to form its belief in the short time frame necessary for high-performance__, and therefore achieves a similar score as the naive population best response baseline. This highlights a key differentiator and source of novelty of TALENTS, which is its ability to perform __intra__-episodic adaptation. This capability is in part made possible by the fixed share tracking regret minimization method (Algorithm 2), and is a reason why TALENTS achieves SOTA performance as a zero-shot coordinator.
>
> ## Switching Between Experts
> Thank you for the question about how TALENTS switches between various experts when observing the player and how experts represent specialization. Fundamentally, our VAE (section 4.1) is used for strategy learning by encoding a set of trajectories from our population agents into a latent space. The optimization target for the VAE is, intuitively, to take a sequence of prior states and actions in order to predict the next actions (e.g. “chop meat”) over some horizon. Given this objective, the latent space can be interpreted as a representation of agent behavior. Clustering this latent space results in a set of experts that exhibit certain behavior patterns, e.g., facilitating a strong preference for cooking rice. We can track the high-level action distribution of each expert to evaluate how specialized each best response behavior is. The mean Jensen-Shannon divergence in the hallway layout across all the action distributions with respect to each other is 0.483 +/- 0.125, indicating moderate behavioral differences. A near-zero divergence would indicate the cooperator has not learned significantly different responses to different partner strategies, while an entirely disjoint near-one divergence would suggest that the cooperator has overfit to each partner type, losing some ability to generalize to out-of-distribution partners.
>
> Exactly quantifying how often TALENTS switches its internal estimate of which expert it is playing with depends on various factors, e.g., how skilled the partner is or how often they change their action patterns. To assess TALENTS’s ability to correctly identify player behaviors, we have conducted additional experiments in which TALENTS is paired with static expert agents generated from its own VAE, which allowed us to know the ground truth clusters the partners are supposed to map to, allowing us to evaluate the player behavior categorization method in isolation. Through this experiment, __TALENTS converged to identifying the correct expert in 91.3% of episodes__. In this setting, it typically took the fixed-share mechanism approximately 30 timesteps to converge to an expert it maintained confidence about for the rest of the episode, seldom changing cluster beliefs partway through an episode. Intuitively, this is not true for experiments with switching policies or dynamic human partners, as we discuss next.
>
> Figure A.6 in the appendix (see supplemental material) provides a figure demonstrating a policy switch, dynamically changing which partner TALENTS is paired with during a game. The experiment shows a partner switch one minute into the game (at time-step 600). As shown in the figure, TALENTS changes the estimate of which partner it is playing with within ~200 timesteps (about 20 seconds), underlining its capability to adjust to changing partners.
>
> ## Impact of Offline Data
> We appreciate the request for analyzing the impact of offline data and agree that this analysis provides valuable insights. To clarify, we generate a dataset from our pretrained behavior policies in order to train the VAE (see Section 4.1). With this data, we ensure that the VAE training is comprehensively covering the possible space of behaviors that potential partners could have. As a preliminary experiment to examine the effect of data quantity, we compare agents trained on a large dataset (550 joint trajectories, 2.64 million total timesteps) with those trained on a smaller dataset (200 joint trajectories, 960,000 total timesteps). We find the large dataset agent to significantly outperform the small dataset agent. Intuitively, the VAE is able to learn a richer latent space with the larger dataset and reduces the risk of overfitting to predominant behaviors.
>
>
> | Model         | Score (± Std. Err.)     |
> |---------------|-------------------------|
> | TALENTS-Large | 834.83 +/- 26.04        |
> | TALENTS-Small | 578.04 +/- 31.78        |
>
> ## RNN Policy:
>
> Thank you for suggesting an additional baseline RNN policy. We agree that such results would be valuable and are providing them below. _TALENTS-Ablated_ refers specifically to a TALENTS agent trained with clustered generative VAE agents (Algorithm 1) but __does not__ employ the fixed-share adaptation mechanism (Algorithm 2), while _TALENTS-Full_ refers to our full method. As can be seen in the following results, the addition of the fixed-share adaptation mechanism (Algorithm 2) provides a significant improvement:
>
> | Model           | Score    |
> |-----------------|----------|
> | TALENTS-Full    | 605.88   |
> | TALENTS-Ablated | 539.71   |
>
> ## Missing caption and incomplete sentence
> Thank you for identifying these issues, we have amended the draft to fix them.
>
> [1] Liang, Y., Chen, D., Gupta, A., Du, S. S., & Jaques, N. (2024). Learning to cooperate with humans using generative agents. Advances in Neural Information Processing Systems, 37, 60061-60087.
>
> [2] Wang, X., Zhang, S., Zhang, W., Dong, W., Chen, J., Wen, Y., & Zhang, W. (2024). Zsc-eval: An evaluation toolkit and benchmark for multi-agent zero-shot coordination. Advances in Neural Information Processing Systems, 37, 47344-47377.
>
> [3] Strouse, D. J., McKee, K., Botvinick, M., Hughes, E., & Everett, R. (2021). Collaborating with humans without human data. Advances in neural information processing systems, 34, 14502-14515.
>
> [4] Zhao, R., Song, J., Yuan, Y., Hu, H., Gao, Y., Wu, Y., ... & Yang, W. (2023, June). Maximum entropy population-based training for zero-shot human-ai coordination. In Proceedings of the AAAI Conference on Artificial Intelligence (Vol. 37, No. 5, pp. 6145-6153).
>
> [5] Zintgraf, L., Devlin, S., Ciosek, K., Whiteson, S., & Hofmann, K. (2021). Deep interactive bayesian reinforcement learning via meta-learning. arXiv preprint arXiv:2101.03864.
>
> [6] Yan, X., Guo, J., Lou, X., Wang, J., Zhang, H., & Du, Y. (2023). An efficient end-to-end training approach for zero-shot human-AI coordination. Advances in neural information processing systems, 36, 2636-2658.
>
> [7] Ma, L., Wang, Y., Zhong, F., Zhu, S. C., & Wang, Y. (2024). Fast peer adaptation with context-aware exploration. arXiv preprint arXiv:2402.02468.
>
> [8] Fu, Haobo, et al. "Greedy when sure and conservative when uncertain about the opponents." International Conference on Machine Learning. PMLR, 2022.

---

> > ### Comment · Reviewer_Eztg · 2025-08-04
> >
> > Thank you the authors for providing the response and additional results. I strongly suggest that the authors include these results into the revised paper. I no longer have any other concerns. I've increased my score to 4 given the current scope and impact of the paper.

---

> > > ### Author Response · Authors · 2025-08-07
> > >
> > > Dear Reviewer Eztg,
> > >
> > > Thank you for your detailed feedback and comments, your time and effort taken to review our submission is greatly appreciated and has been a great help. We have revised our submission to include the additional baselines, ablations, and discussions we presented.
> > >
> > > Kind regards,
> > > The Authors

---

### Official Review · Reviewer_mm8i · 2025-07-03

**Clarity:** 3
**Significance:** 2
**Originality:** 2
**Rating:** 3
**Confidence:** 4

**Summary:**

The paper introduces **TALENTS**, a framework for zero-shot coordination in cooperative multi-agent settings by learning a latent strategy space via a variational autoencoder, clustering behaviors into discrete types, and training a strategy-conditioned cooperator agent. Online adaptation to novel partners is achieved through a fixed-share regret-minimization algorithm that dynamically infers and switches between latent strategy experts during interaction.

However, **the core innovation**—combining a VAE-based embedding with clustering and regret minimization—**offers only an incremental advance** over existing latent-embedding and online adaptation methods, and **the experimental design** (single benchmark domain, limited baseline set, and hand-tuned cluster counts) **is insufficiently comprehensive to meet NeurIPS standards**.

**Questions:**

**Questions for the Authors**

1. **Cluster sensitivity**: How does performance vary with different choices of latent dimensionality and cluster count? Can you provide an ablation?

2. **Cross-domain generality**: Can TALENTS be applied to other cooperative tasks (e.g., Hanabi, pursuit–evasion)? If not, what modifications would be needed?

3. **Real-time feasibility**: What is the per-step inference time for strategy inference and cooperator action selection, and how does this scale with the number of clusters?

**Ethical Concerns:**

["NO or VERY MINOR ethics concerns only"]

**Final Justification:**

After discussion with the authors, most of my concerns have been addressed, but I still believe the paper lacks sufficient novelty.

**Limitations:**

yes

**Paper Formatting Concerns:**

No paper formatting concerns.

**Quality:**

3

**Strengths And Weaknesses:**

Strengths:
1. **Clear technical exposition**: The methodology (VAE training, K-means clustering, strategy-conditioned PPO, fixed-share adaptation) is described in a coherent, step-by-step manner.

2. **Human-agent evaluation**: Inclusion of a human subject study with statistical tests (mixed ANOVA, p-values) adds practical validation beyond purely simulated benchmarks.

3. **Implementation details**: The manuscript provides computational resource specifications and training times, aiding reproducibility.

Weaknesses:
1. **Incremental novelty**: Key components (latent strategy embeddings, clustering, expert weighting) closely mirror prior work (e.g., GAMMA, MeLIBA) without a fundamentally new algorithmic insight.

2. **Limited baselines**: Absence of comparisons to stronger or more diverse adaptation frameworks (e.g., Bayesian meta-task reuse) weakens claims of state-of-the-art performance.

3. **Single-domain focus**: All experiments are confined to Overcooked; generalization to other collaborative tasks is untested.

4. **Cluster selection**: Reliance on K-means with silhouette analysis lacks robustness; no sensitivity analysis over the number of clusters or VAE latent size is provided.

5. **Computational cost**: While training times are given, there is no discussion of inference latency, which is critical for real-time human–agent interaction.

---

> ### Author Rebuttal · Authors · 2025-07-31
>
> We thank the reviewer for the thoughtful and detailed review. We appreciate the informed feedback and suggested areas for improvement. We appreciate the positive remarks regarding our __clear methodology__ discussion and the __strength of our human subject study__ which lends credibility to the efficacy of the TALENTS method. Furthermore, we are encouraged by the reviewer’s assessment that our specification of compute requirements for training __aid the reproducibility of our method__. Below, we will address the stated weaknesses and questions.
>
> ## Incremental novelty:
> Our work is inspired by the idea of latent policy representations and online partner adaptation. However, there are two primary novelties of TALENTS which significantly differentiate the method from existing state-of-the-art methods:
>
> 1. Learning a _clustered latent embedding of agent behaviors_ from offline game trajectories to both condition a cooperator agent and serve as a generative model to produce strategy-specific partner agents during training. GAMMA [1] introduced the method of using a variational autoencoder to create a latent representation of agents from offline trajectories. However, we extend GAMMA by adding a recurrent element to the decoder architecture, enabling the latent to encode a _longer-term behavioral representation of agents_. Furthermore, we propose silhouette analysis as a method of partitioning the learned latent into individual clusters, enabling our agent to more efficiently learn the best response behavior to rarer agent strategies (in GAMMA, these strategies are sampled much less frequently compared to the predominant strategy). Lastly, the TALENTS cooperator is _explicitly conditioned on these clusters_ during training, which is a novel contribution with respect to generative agent-trained cooperators.
>
> 2. TALENTS enables __intra__-episodic adaptation to zero-shot coordination problems through a tracking regret minimization framework. Prior work in the agent modeling and online adaptation setting (MeLIBA, GSCU, etc) define the adaptation objective as identifying and optimizing strategies over multiple episodes. However, in the zero-shot coordination setting, successful agents must be able to adapt to unseen partner agents within one episode. TALENTS proposes fixed-share tracking regret minimization as an effective method for accomplishing this, and demonstrates in an ablation (Fig 3) the necessity of optimizing tracking regret as opposed to static regret. This novel contribution results in a SOTA adaptive zero-shot coordination agent.
>
> ## Limited Baselines:
> We appreciate the reviewer for pointing out the need for additional baselines and are excited to report that we have conducted additional experiments with two more baselines, namely E3T and PACE. We have added these baselines in addition to our prior evaluation of TALENTS against several leading zero-shot coordination baselines (GAMMA, MEP, FCP, BP). E3T [2] utilizes a modified self-play objective alongside a partner modeling module to perform ZSC while PACE [3] adapts its behavior based on a context encoder. Preliminary results are as follows:
>
> | Model   | Score (± Std. Err.)     |
> |---------|-------------------------|
> | E3T     | 548.66 ± 25.44          |
> | PACE    | 313.34 ± 18.48          |
> | GAMMA   | 374.10 ± 24.99          |
> | BR      | 343.71 ± 24.27          |
> | TALENTS | 605.88 ± 27.89          |
>
> TALENTS remains the top performer out of all the assessed methods, while E3T is able to outperform all the other baselines. PACE underperforms, achieving comparable scores to the population best response baseline. This can be attributed to the fact that PACE, alongside similar methods like MeLIBA [4], are effective __inter__-episodic adaptation frameworks. However, in this zero-shot coordination problem setting, PACE is unable to form its belief in the short time frame necessary for high-performance, and therefore achieves a similar score as the naive population best response baseline. This highlights a key differentiator and source of novelty of TALENTS, which is its ability to perform __intra__-episodic adaptation.
>
> We also appreciate the suggestion to consider “Bayesian meta-task reuse”. Unfortunately, it is unclear which specific method the reviewer is referring to. Thus, we would be grateful if the reviewer could kindly point us to a relevant reference or paper, so that we can incorporate that work.
>
> ## Cluster Selection and Sensitivity:
> We appreciate the request for further insights into the number of clusters as well as the sensitivity of TALENTS with respect to the number of chosen clusters. To clarify, we utilize silhouette analysis to cluster our latent embedding space with the goal of finding a suitable representation of discrete strategy types. There is precedence of this method as an effective way of partitioning a latent embedding space [5,6]. To the reviewer’s point, prior work in opponent modeling and latent encoding methods have explored modifications to the VAE objective in order to learn a clustered embedding space in a more end-to-end fashion [9,4], but these objectives often require privileged information in order to function, such as the identity of the agent from which the trajectory originated from [9]. Unlike these methods, TALENTS does not make such assumptions. To evaluate the efficacy of silhouette analysis for selecting an optimal number of clusters, we perform a sensitivity analysis over a range of clusters, ranging from the naive one-cluster case to 16 clusters. We find that the silhouette analysis optimal number of clusters (11) achieves the highest score, empirically demonstrating the strength of our clustering approach.
>
> | Number of Clusters   | Score (± Std. Err.)     |
> |---------|-------------------------|
> | 1     | 650.83 ± 28.00          |
> | 2    | 701.68 ± 11.93         |
> | 4   | 620.35 ± 39.01          |
> | 8      | 740.93 ± 79.15          |
> | 11 | 865.82 ± 15.11          |
> | 14 | 724.94 ± 40.60          |
> | 16 | 648.57 ± 26.26          |
>
> ## Cross-domain generality:
> Thank you for requesting additional information with respect to the applicability of TALENTS to additional environments, such as Hanabi or pursuit-evasion tasks. TALENTS, as proposed in this work, is targeting pure collaborative tasks between multiple players in a fully observable environment. Overcooked is a challenging and complex environment for human-agent, as well as agent-agent teamwork, requiring effective collaboration between all players [1,7,8]. As such, Hanabi, which utilizes a partial information setting, and pursuit-evasion games, which utilize an adversarial component, are out of scope for the proposed method, but provide an interesting avenue for TALENTS in potential future work.
>
> ## Computational Cost and Real-Time Capabilities
>
> We acknowledge the need for further clarification of TALENTS’s ability to run in real-time and perform inference with minimal latency to collaborate effectively. As such, we measured the inference time of TALENTS. We tracked the average total time taken from when the policy receives an observation from the environment to when it sends its action back to the environment (expert evaluation and cluster inference occur between those two states). We compare to a policy without any adaptation component as a baseline.
>
> | Method   | Clusters | Mean Latency |
> |----------|----------|---------|
> | TALENTS  | 4        | 8.7 ms  |
> | TALENTS  | 11       | 18.8 ms |
> | TALENTS  | 15       | 23 ms   |
> | Baseline | N/A      | 7.5 ms  |
>
> Human-agent experiments in the Overcooked environment run at 10 frames per second, so even with a 23ms latency and the extreme case of expert computation every frame, there is still a considerable margin. The inference time roughly scales linearly with respect to the cluster number. If the number of experts was considerably larger, the cluster inference steps could be parallelized across GPUs, leading to decreased latency. However, since this latency was not an issue in our experiments, we saw no need to make this optimization.
>
> [1] Liang, Y., Chen, D., Gupta, A., Du, S. S., & Jaques, N. (2024). Learning to cooperate with humans using generative agents. Advances in Neural Information Processing Systems, 37, 60061-60087.
>
> [2] Yan, X., Guo, J., Lou, X., Wang, J., Zhang, H., & Du, Y. (2023). An efficient end-to-end training approach for zero-shot human-AI coordination. Advances in neural information processing systems, 36, 2636-2658.
>
> [3] Ma, L., Wang, Y., Zhong, F., Zhu, S. C., & Wang, Y. (2024). Fast peer adaptation with context-aware exploration. arXiv preprint arXiv:2402.02468.
>
> [4] Zintgraf, L., Devlin, S., Ciosek, K., Whiteson, S., & Hofmann, K. (2021). Deep interactive bayesian reinforcement learning via meta-learning. arXiv preprint arXiv:2101.03864.
>
> [5] Zhao, M., Simmons, R., & Admoni, H. (2022, October). Coordination with humans via strategy matching. In 2022 IEEE/RSJ International Conference on Intelligent Robots and Systems (IROS) (pp. 9116-9123). IEEE.
>
> [6] Pandya, R., Zhao, M., Liu, C., Simmons, R., & Admoni, H. (2024, May). Multi-agent strategy explanations for human-robot collaboration. In 2024 IEEE International Conference on Robotics and Automation (ICRA) (pp. 17351-17357). IEEE.
>
> [7] Zhao, R., Song, J., Yuan, Y., Hu, H., Gao, Y., Wu, Y., ... & Yang, W. (2023, June). Maximum entropy population-based training for zero-shot human-ai coordination. In Proceedings of the AAAI Conference on Artificial Intelligence (Vol. 37, No. 5, pp. 6145-6153).
>
> [8] Carroll, M., Shah, R., Ho, M. K., Griffiths, T., Seshia, S., Abbeel, P., & Dragan, A. (2019). On the utility of learning about humans for human-ai coordination. Advances in neural information processing systems, 32.
>
> [9] Papoudakis, G., & Albrecht, S. V. (2020). Variational autoencoders for opponent modeling in multi-agent systems. arXiv preprint arXiv:2001.10829.

---

> > ### Author Response · Authors · 2025-08-06
> >
> > Dear Reviewer mm8i,
> >
> > Thank you for your thoughtful and thorough review of our submission, we greatly appreciate the time you dedicated to evaluating our work. We have carefully considered each of your comments and have prepared a detailed rebuttal that addresses your questions and comments. We believe our responses provide the clarifications needed to resolve your concerns and strengthen our manuscript.
> >
> > Should you require any additional clarification or have further questions, we would be happy to provide the necessary details. We also welcome any additional suggestions you might have for enhancing the quality of our work.
> > We hope our comprehensive response has adequately addressed your concerns and would be grateful for your updated evaluation of our submission.
> >
> > Kind regards,
> >
> > The Authors

---

### Note · Authors · 2025-08-14

We thank all the reviewers and AC for their time and detailed feedback provided throughout this review process. The questions and comments raised by the reviewers have led us to clarify key parts of our approach through additional experiments and ablations, and have provided an improved understanding of the strengths and performance of our zero-shot coordination method.

We appreciate the positive remarks reviewers had about our work, including that it is __well-motivated, grounded in principled approaches__, and shows effective performance in experiments, which includes a __comprehensive human subject study__.
Based on the valuable suggestions provided, we included additional clarifications and experiments to assess our method more comprehensively. To this end, we implemented __two additional baselines, E3T and PACE__, and ran preliminary evaluations. E3T, a SOTA self-play method with a partner modeling module, serves as our strongest baseline; however, TALENTS performs statistically significantly better than even this SOTA method. PACE, a SOTA population-based few-shot adaptation method, on the other hand, was unable to adapt fast enough to perform effectively in our zero-shot coordination setting, highlighting TALENTS’ strength as an __intra-episodic cooperator agent__.

To evaluate the efficacy of our regret-minimized partner module, we performed an additional ablation in which we used Algorithm 1 to train an RNN-policy, finding the ablated version to have an __11% performance drop__ in the open and hallway layouts.
Addressing reviewer questions about silhouette analysis for determining latent space cluster counts, we ran a __sensitivity analysis__ over cluster numbers in the open layout, finding empirically that
__silhouette-optimal clusters yielded policies with the highest performance__. We also found that inference time scales linearly with the number of clusters queried at runtime, though these operations could hypothetically be parallelized to save time. In our experiments, our highest-latency agent (23ms on 2x AMD EPYC 7713 64-core processors, 5x Nvidia RTX 6000 Ada GPUs) remained under the experiment’s required 100ms response time, allowing __seamless real-time human-agent gameplay__.

Overall, we have received invaluable feedback throughout this review process and have __incorporated all additional experiments and clarifications into the manuscript__. We thank the reviewers and AC for their time and expertise in reviewing our work.

---

### Decision · Program_Chairs · 2025-09-17

**Decision:**

Accept (poster)

**Comment:**

The paper proposes a novel method for training an adaptive cooperative agent that explicitly adapt online by adjusting its internal strategy clusters. The method first learns a VAE from offline joint trajectories data. Then apply K-mean clustering on the VAE's latent space. The cooperative agent is then trained by sampling "partners" from the VAE clusters. During test-time, the "Fixed-Share" is used by to infer the current cluster of the partner, which overall gives the adaptivity to the cooperative agent. The method provide good empirical results across Overcooked layouts.

Most reviewers appreciate the novelty of the new method and human studies. While there are some concerns about sensitivity of clustering and insufficient baselines. During the rebuttal, the authors addressed most concerns which are also acknowledged by the reviewers. The AC believes this paper makes a valuable contribution to the community and thus recommends acceptance.